# How clustered protocadherin binding specificity is tuned for neuronal self-/nonself-recognition

**Kerry Marie Goodman[1†], Phinikoula S Katsamba[1†], Rotem Rubinstein[2,3], Göran Ahlsén[1], Fabiana Bahna[1], Seetha Mannepalli[1], Hanbin Dan[4], Rosemary V Sampogna[4], Lawrence Shapiro[1,5]\*, Barry Honig[1,4,5,6]\***

[1]Zuckerman Mind, Brain and Behavior Institute, Columbia University, New York, United States; [2]School of Neurobiology, Biochemistry and Biophysics, Tel Aviv University, Tel Aviv, Israel; [3]Sagol School of Neuroscience, Tel Aviv University, Tel Aviv, Israel; [4]Department of Medicine, Division of Nephrology, Columbia University, New York, United States; [5]Department of Biochemistry and Molecular Biophysics, Columbia University, New York, United States; [6]Department of Systems Biology, Columbia University, New York, United States

**\*For correspondence:**
lss8@columbia.edu (LS);
bh6@cumc.columbia.edu (BH)

[†]These authors contributed equally to this work

**Competing interest:** The authors declare that no competing interests exist.

**Abstract** The stochastic expression of fewer than 60 clustered protocadherin (cPcdh) isoforms provides diverse identities to individual vertebrate neurons and a molecular basis for self-/nonself-discrimination. cPcdhs form chains mediated by alternating *cis* and *trans* interactions between apposed membranes, which has been suggested to signal self-recognition. Such a mechanism requires that cPcdh *cis* dimers form promiscuously to generate diverse recognition units, and that *trans* interactions have precise specificity so that isoform mismatches terminate chain growth. However, the extent to which cPcdh interactions fulfill these requirements has not been definitively demonstrated. Here, we report biophysical experiments showing that cPcdh *cis* interactions are promiscuous, but with preferences favoring formation of heterologous *cis* dimers. *Trans* homophilic interactions are remarkably precise, with no evidence for heterophilic interactions between different isoforms. A new C-type cPcdh crystal structure and mutagenesis data help to explain these observations. Overall, the interaction characteristics we report for cPcdhs help explain their function in neuronal self-/nonself-discrimination.

## Editor's evaluation

The direct investigation of homotypic and heterotypical preference between cis and trans interactions among the protocadherin isoforms is an important step to understand the mechanisms of self avoidance. We are particularly excited about the discovery that the discovery that showed cis interactions are promiscuous, but with preferences favoring formation of heterologous cis dimers. Trans-homophilic interactions are remarkably precise, with no evidence for heterophilic interactions between different isoforms.

## Introduction

Clustered protocadherins (cPcdhs) are a large family of cadherin-like proteins named for the clustered arrangement of their genes in vertebrate genomes (*Wu and Maniatis, 1999*; *Wu et al., 2001*). cPcdhs

**Figure 1.** Clustered protocadherin (cPcdh) domain organization and extracellular interactions. (**A**) Schematic depicting the domain organization of cPcdhs. EC, extracellular cadherin domain; TM, transmembrane domain; ECD, ectodomain; ICD, intracellular domain. (**B**) Schematic of two cPcdhs interacting via the EC1–4 *trans* interface. (**C**) Schematic of two cPcdhs interacting via the EC5–6/EC6 *cis* interface. (**D**) Schematic depiction of the *cis/trans* cPcdh zipper comprising multiple cPcdh isoforms (various colors) engaged in homophilic *trans* interactions and promiscuous *cis* interactions as required for the proposed 'isoform-mismatch chain-termination model' of cPcdh-mediated neuronal self-recognition and self-avoidance.

play roles in many facets of neural development (**Peek et al., 2017**), including circuit development, most notably neurite self-avoidance in vertebrates (**Kostadinov and Sanes, 2015**; **Lefebvre et al., 2012**; **Mountoufaris et al., 2017**), and tiling (**Chen et al., 2017**). In self-avoidance, neurites from the same neuron (sister neurites) actively avoid one another, whereas neurons from different neurons can freely interact. Tiling is similar to self-avoidance, but in tiling all neurons acquire the same identity, so that there is uniform repulsion among self- and nonself-neurites (**Chen et al., 2017**). Self-avoidance among sister neurites leads to the characteristic arbor structures of dendritic trees, and prevents the formation of self-synapses (**Kostadinov and Sanes, 2015**; **Lefebvre et al., 2012**).

The molecular mechanisms through which neurons discriminate self from nonself, differ between vertebrate and most invertebrate animals. For arthropod invertebrates such as *Drosophila melanogaster*, self-avoidance is mediated by immunoglobulin superfamily Dscam1 cell surface proteins. The stochastic alternative splicing of *Dscam1* pre-mRNAs can, in principle, generate 19,008 distinct extracellular isoforms; the vast majority of which, based on ELISA-based binding assays, mediate homophilic recognition (**Miura et al., 2013**; **Schmucker et al., 2000**; **Wojtowicz et al., 2004**; **Wojtowicz et al., 2007**). Each *Drosophila* neuron expresses a repertoire estimated at 10–50 isoforms and the large number of Dscam1 isoforms ensures a low probability that any two contacting neurons will have an identical or even a similar isoform repertoire thus minimizing the chance of inappropriate avoidance between nonself-neurons (**Hattori et al., 2009**).

In mammalian nervous systems, cPcdh isoform expression is controlled by the unique organization of three tandem gene clusters, *Pcdhα*, *Pcdhβ*, and *Pcdhγ* (**Wu and Maniatis, 1999**), with each cluster containing multiple variable exons, which encode full cPcdh ectodomain regions with six extracellular cadherin (EC) domains, a single transmembrane region, and a short cytoplasmic extension (**Figure 1A**). The *Pcdhα* and *Pcdhγ* gene clusters also contain three 'constant' exons that encode cluster-specific intracellular domains. The last two variable exons in the *Pcdhα* gene cluster and the last three variable exons of the *Pcdhγ* gene cluster diverge in sequence from other cPcdh isoforms and are referred to as 'C-type' cPcdhs (**Wu and Maniatis, 1999**; **Wu et al., 2001**). Sequence differences further subdivide *Pcdhγ* genes into two subfamilies – *PcdhγA* and *PcdhγB* (**Wu and Maniatis, 1999**). The full mouse cPcdh complement is comprised of 53 non-C-type cPcdhs, commonly known as alternate cPcdhs (α1–12, β1–22, γA1–12, and γB1–7), whose expression choices vary stochastically between cells through alternate promoter choice (**Canzio and Maniatis, 2019**) and 5 C-type cPcdhs (αC1, αC2, γC3, γC4, and γC5), which are constitutively expressed. cPcdh expression, either stochastic or constitutive, varies between cell types: for example, olfactory sensory neurons express ~5–10 cPcdhs stochastically; Purkinje neurons express ~10 alternate cPcdhs stochastically and all five C-types constitutively (**Esumi et al., 2005**; **Kaneko et al., 2006**); and serotonergic neurons express just αC2 constitutively (**Canzio and Maniatis, 2019**; **Chen et al., 2017**). While the cPcdh and Dscam1

systems bear striking similarities, the relatively small number of cPcdh isoforms – fewer than 60 – has presented a significant challenge to generation of sufficient diversity to provide mammalian neurons with functionally unique identities.

Solution biophysics and functional mutagenesis studies have shown that cPcdhs interact in *trans* through antiparallel interactions between their EC1–4 regions (*Rubinstein et al., 2015*), and crystal structures of alternate α, β, and γ cPcdh *trans* homodimers have revealed interfaces involving EC1 interacting with EC4 and EC2 with EC3 (*Figure 1B*; *Goodman et al., 2016c*; *Goodman et al., 2016a*; *Nicoludis et al., 2016*; *Rubinstein et al., 2015*; *Thu et al., 2014*). cPcdhs also form *cis* dimers through their membrane-proximal EC5–6 regions, and are presented on cell surfaces as *cis* dimers (*Goodman et al., 2017*; *Rubinstein et al., 2015*; *Thu et al., 2014*). Crystal structures of *cis*-interacting protocadherin ectodomains (*Brasch et al., 2019*; *Goodman et al., 2017*) have revealed an asymmetrical interaction mode, where one molecule interacts through elements of EC5 and EC6, and the other interacts exclusively through EC6 (*Figure 1C*). To date, structural studies of C-type cPcdh interactions have not been available. Here, we extend our molecular understanding of cPcdhs to C-type isoforms as well, with the goal of understanding the evolutionary design of the entire family.

In order to explain how about 60 cPcdh isoforms can provide a comparable or even greater level of neuronal diversity as 19,000 Dscam isoforms, *Rubinstein et al., 2015* proposed that cPcdhs located on apposed membrane surfaces would form an extended zipper-like lattice through alternating *cis* and *trans* interactions (*Figure 1D*). In self-interactions – between two membranes with identical cPcdh repertoires – these chains would grow to form large structures, limited mainly by the number of molecules (*Brasch et al., 2019*; *Rubinstein et al., 2015*). However, in nonself-interactions – between two membranes with differing cPcdh repertoires – such large linear assemblies would not form since even a single mismatch between expressed isoforms would terminate chain assembly (*Brasch et al., 2019*; *Rubinstein et al., 2017*; *Rubinstein et al., 2015*). This 'isoform-mismatch chain-termination model' for the 'barcoding' of vertebrate neurons envisions the assembly of long cPcdh chains between sites of neurite–neurite contact to represent the signature of 'self', which is then translated by downstream signaling that leads to self-avoidance (*Fan et al., 2018*). X-ray crystallographic studies and cryo-electron tomography studies of the full-length cPcdh ectodomains bound between the surfaces of adherent liposomes revealed the existence of linear zippers thus providing strong evidence supporting the validity of the model (*Brasch et al., 2019*). However, crucial questions remain unanswered. Here, a number of them are addressed.

1. For the proposed mechanism to successfully explain neuronal barcoding, *cis* interactions must be promiscuous to generate diverse repertoires of *cis*-dimeric biantennary 'interaction units', while *trans* interactions must be highly specific so that mismatched isoforms do not inappropriately enable growth of the chain through heterophilic interactions. While cell aggregation assays have suggested *trans* homophilic specificity, these assays only reflect a *competition* between different cell populations and thus do not inform as to the strength of heterophilic interactions. Moreover, the results of cell aggregation assays depend critically on the *relative* strengths of homophilic and heterophilic interactions and thus do not inform as to actual binding affinities (*Honig and Shapiro, 2020*). It is thus necessary to establish the extent to which heterophilic *trans* interactions are truly disallowed.

2. The assumption that *cis* interactions are promiscuous is based in large part on the fact that α-cPcdhs and γC4 cannot reach the cell surface without binding in *cis* to another 'carrier' isoform (*Bonn et al., 2007*; *Goodman et al., 2016a*; *Murata et al., 2004*; *Schreiner and Weiner, 2010*; *Thu et al., 2014*). As is the case for *trans* interactions, the strength of *cis* interactions has only been probed quantitatively in a small number of cases so that the term 'promiscuous' is qualitative at best. In fact, as compared to γB and β cPcdh isoforms, most γA-Pcdhs do not form measurable *cis* homodimers in solution (*Goodman et al., 2016a*; *Figure 4—source data 1*). Nevertheless, all γA-Pcdhs are still able to reach the cell surface when expressed alone (*Thu et al., 2014*). This observation can be understood if the *cis* dimerization affinity of γA-Pcdhs is large enough to enable them to dimerize in the 2D membrane environment (*Goodman et al., 2016a*; *Wu et al., 2013*). Nevertheless, their weak dimerization affinities suggest, more generally, that cPcdhs may exhibit a range of *cis* dimerization affinities. We establish below that a wide range of affinities does in fact exist and, strikingly, most homophilic *cis* interactions are weaker than their heterophilic counterparts. We consider the functional implications of this novel observation in the discussion.

3. Structures have not yet been determined for complete C-type cPcdh ectodomains. Yet these isoforms play unique functional roles, some of which have no apparent connection to isoform diversity. For example, a single C-type isoform is sufficient for tiling which can be simply understood in terms of the formation of zippers containing identical homodimers so that all interacting neurons will avoid one another (*Chen et al., 2017*). Moreover, Garrett et al. discovered that neuronal survival and postnatal viability are controlled solely by γC4 suggesting a function that is unique to this isoform (although it presumably requires β and/or other γ carriers to reach the cell surface) (*Garrett et al., 2019*). Additionally, a recent paper by Iqbal et al. has shown that genetic γC4 variants cause a neurodevelopmental disorder which is potentially linked to γC4's role in programmed cell death of neuronal cells (*Iqbal et al., 2021*). Below we report extensive biophysical interaction studies of C-type isoform ectodomains and report the first crystal structure of a *trans* dimer formed by γC4. Our findings reveal that the specialized functions of C-type cPcdhs probably do not involve unique structural or biophysical properties of their ectodomains.

Overall, in accordance with the requirements of the isoform-mismatch chain-termination model, we find that *trans* homophilic interactions are remarkably precise, with no evidence for heterophilic interactions between different cPcdh isoforms. In contrast cPcdh *cis* interactions are largely promiscuous but with relatively weak intrasubfamily and, especially, homophilic interactions. Possible implications of this somewhat surprising finding are considered in the discussion. Our study reveals how the extraordinary demands posed by the need to assign each neuron with a unique identity are met by an unprecedented level of protein–protein interaction specificity.

## Results
### cPcdh *trans* interactions are strictly homophilic

We generated biotinylated ectodomain fragments containing the *trans*-interacting EC1–4 regions (*Nicoludis et al., 2015*; *Rubinstein et al., 2015*) of six representative α, β, γA, and γB mouse cPcdh isoforms – α7, β6, β8, γA8, γA9, and γB2 – which include the most closely related isoforms by sequence identity from the β and γA subfamilies (β6/8 and γA8/9) (*Rubinstein et al., 2015*). These molecules were coupled over independent NeutrAvidin-immobilized flow cells and *trans*-interacting ectodomain fragments of multiple members of each mouse cPcdh subfamily, including the C-types (α4, α7, α12, β6, β8, γA4, γA8, γA9, γB2, γB4, γB5, αC2, γC3, γC4, and γC5), were then flowed over the six cPcdh surfaces to assess their binding. The surface plasmon resonance (SPR)-binding profiles reveal strictly homophilic binding (*Figure 2A*). All ectodomain fragments used in these SPR experiments were confirmed to form homodimers in solution by sedimentation equilibrium analytical ultracentrifugation (AUC) (*Figure 2—source data 1*), validating that these proteins are well behaved and active. Remarkably, no heterophilic binding was observed for any of the analytes over any of the six surfaces (*Figure 2A*). Even β6/8 and γA8/9 that have 92% and 82% sequence identities, respectively, in their *trans*-binding EC1–4 regions exhibit no heterophilic binding. We estimate that, for heterophilic *trans* dimers, the lower limit for the dissociation constant ($K_D$) would be ~200 µM. Mutations designed to disrupt α7, β6, and γA8 *trans* interaction inhibited homophilic binding, demonstrating that the observed binding occurs via the *trans* interface (*Figure 2—figure supplement 1A*; *Goodman et al., 2016a*; *Goodman et al., 2016c*; *Rubinstein et al., 2015*). This behavior is unlike that of other adhesion receptor families where, whether they display homophilic or heterophilic preferences, the signal is never as binary as the one shown in *Figure 2* (*Honig and Shapiro, 2020*).

Much of the original evidence as to homophilic specificity was based on cell aggregation assays (*Rubinstein et al., 2015*; *Schreiner and Weiner, 2010*; *Thu et al., 2014*) and it is of interest to compare the results obtained from these assays to those obtained from SPR. We do this in the context of examining the heterophilic binding specificity between β6$_{1-4}$ and β8$_{1-4}$ *trans* fragments that share 92% sequence identity and differ at only five residues (*Figure 2—figure supplement 2A*), within their respective binding interfaces (*Goodman et al., 2016c*). Each of these residues was mutated individually and in combination. *Figure 2—figure supplement 2B, C* displays SPR profiles and cell aggregation images, respectively, for wild-type β6 and β8 and for the various mutations. We first note that changing all five residues in β6 to those of β8 generates a mutant protein with essentially wild-type β8 properties; it binds strongly to β8 but not to β6 as seen in SPR and also forms mixed aggregates with β8 but not β6. In contrast, most of the single residue mutants retain β6-like properties in both assays whereas double and triple mutants exhibit intermediate behavior between β6 and β8. These

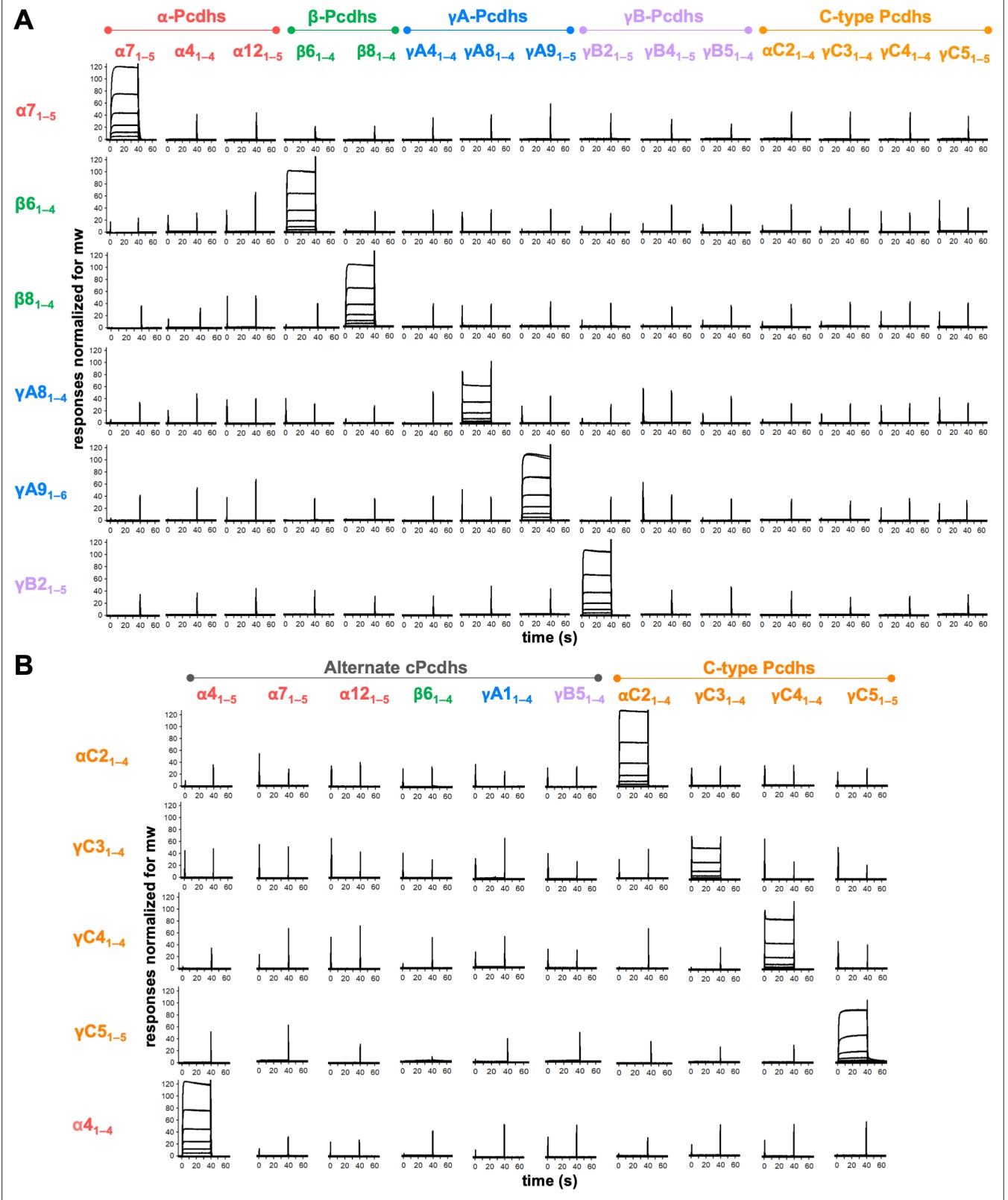

**Figure 2.** Clustered protocadherins (cPcdhs) show strict homophilic specificity in their *trans* interactions. (**A**) Surface plasmon resonance (SPR) binding profiles of cPcdh *trans* fragment analytes from all cPcdh subfamilies (denoted in the top row) flowed over six surfaces coated with alternate cPcdh *trans* fragments (rows). Responses over all surfaces are drawn on the same scale and normalized for molecular weight (mw). (**B**) SPR binding profiles of cPcdh

*Figure 2 continued on next page*

Figure 2 continued

*trans* fragment analytes from all cPcdh subfamilies (shown in columns) flowed over individual surfaces coated with C-type and α4 cPcdh *trans* fragments (rows). Responses over all surfaces are drawn on the same scale and normalized for molecular weight.

The online version of this article includes the following source data and figure supplement(s) for figure 2:

**Source data 1.** Sedimentation equilibrium analytical ultracentrifugation data for trans SPR reagents.

**Figure supplement 1.** *Trans* interface mutants demonstrate homophilic interactions observed in surface plasmon resonance (SPR) are mediated by the *trans* dimer interface.

**Figure supplement 2.** Mutagenesis experiments reveal role in *trans* specificity for the five interfacial residue differences between close pair $\beta6_{1-4}$ and $\beta8_{1-4}$.

results demonstrate that despite the 92% sequence identity between β6 and β8, their highly specific homophilic properties can be attributed to five interfacial residues. Moreover, the cell aggregation assays are consistent with the heterophilic binding traces measured by SPR; cells expressing mutants that generate strong SPR signals with either wild-type β6 or β8 also form mixed aggregates with cells expressing the same wild-type protein.

Of note, *trans*-interacting fragments of all four C-type cPcdhs tested showed no binding over the alternate cPcdh SPR surfaces (**Figure 2A**). To test whether C-type cPcdhs also show strict homophilic specificity with respect to each other we coupled biotinylated *trans*-interacting fragments of αC2, γC3, γC4, and γC5 to SPR chips and passed the same four fragments alongside alternate cPcdh *trans* fragments over these four surfaces. Only homophilic binding was observed, with each of the four C-type fragments binding to its cognate partner and no other isoform (**Figure 2B**). Disrupting the γC5 *trans* interaction with the S116R mutation (**Rubinstein et al., 2015**), inhibited binding to the γC5 surface, demonstrating that the observed binding occurs via the *trans* interface (**Figure 2—figure supplement 1B**).

In contrast to the other C-type isoforms, αC1 does not mediate cell–cell interactions in cell aggregation assays even when coexpressed with cPcdhs that facilitate cell-surface delivery of γC4 (**Thu et al., 2014**). Although we have been able to produce an αC1 EC1–4 fragment the recombinant molecule forms disulfide-linked multimers which are likely nonnative, precluding confident examination of αC1's potential *trans* interactions. Notably, the sequence of mouse αC1 reveals the EC3:EC4 linker does not contain the full complement of calcium-coordinating residues, which may impact the structure and binding properties of this protein (**Thu et al., 2014**).

Since all the cPcdh *trans* fragment molecules used in these SPR experiments homodimerize our SPR data cannot be used to determine accurate binding affinities (**Rich and Myszka, 2007**). We therefore used AUC to measure the *trans* homodimer $K_D$s (**Figure 2—source data 1**) revealing a >200-fold range of binding affinities, from 2.9 μM ($\alpha7_{1-5}$) to >500 μM ($\gamma C4_{1-4}$). Regardless of their *trans*-binding affinity, all cPcdhs (except αC1) have previously been shown to effectively mediate cell–cell interactions in cell aggregation assays (**Schreiner and Weiner, 2010**; **Thu et al., 2014**).

## Crystal structure of C-type cPcdh γC4 reveals EC1–4-mediated head-to-tail *trans* dimer interaction

The biophysical properties of C-type cPcdhs pose a number of interesting questions: Despite their more divergent sequences compared with alternate cPcdhs, AUC data have confirmed that C-type cPcdhs αC2, γC3, and γC5 form *trans* dimers using their EC1–4 domains (**Goodman et al., 2016a**; **Rubinstein et al., 2015**). However, $\gamma C4_{1-4}$ behaved as a very weak dimer in AUC ($K_D$ >500 μM; **Figure 2—source data 1**), nevertheless full-length γC4 can mediate cell aggregation when delivered to the cell surface by coexpression with a 'carrier' cPcdh (**Thu et al., 2014**). In addition, C-type isoforms have unique expression profile and function compared to alternate cPcdhs (**Canzio and Maniatis, 2019**; **Mountoufaris et al., 2018**). However, there are no published crystal structures of C-type cPcdh *trans* dimers. We therefore sought to crystallize a mouse C-type cPcdh engaged in a *trans* interaction and obtained two distinct crystal forms of $\gamma C4_{EC1-4}$, one at 2.4 Å resolution (crystallized at pH 7.5) and the other with anisotropic diffraction at 4.6/3.9/3.5 Å resolution (**Figure 3A**, **Figure 3—figure supplement 1A, B**, **Figure 3—source data 1**) (crystallized at pH 6.0). Both crystal structures revealed an EC1–4-mediated head-to-tail *trans* dimer: The 4.6/3.9/3.5 Å crystal structure appears to have a fully intact *trans* interface with a total buried surface area of 3800 Å², which is a similar size to other cPcdh *trans*

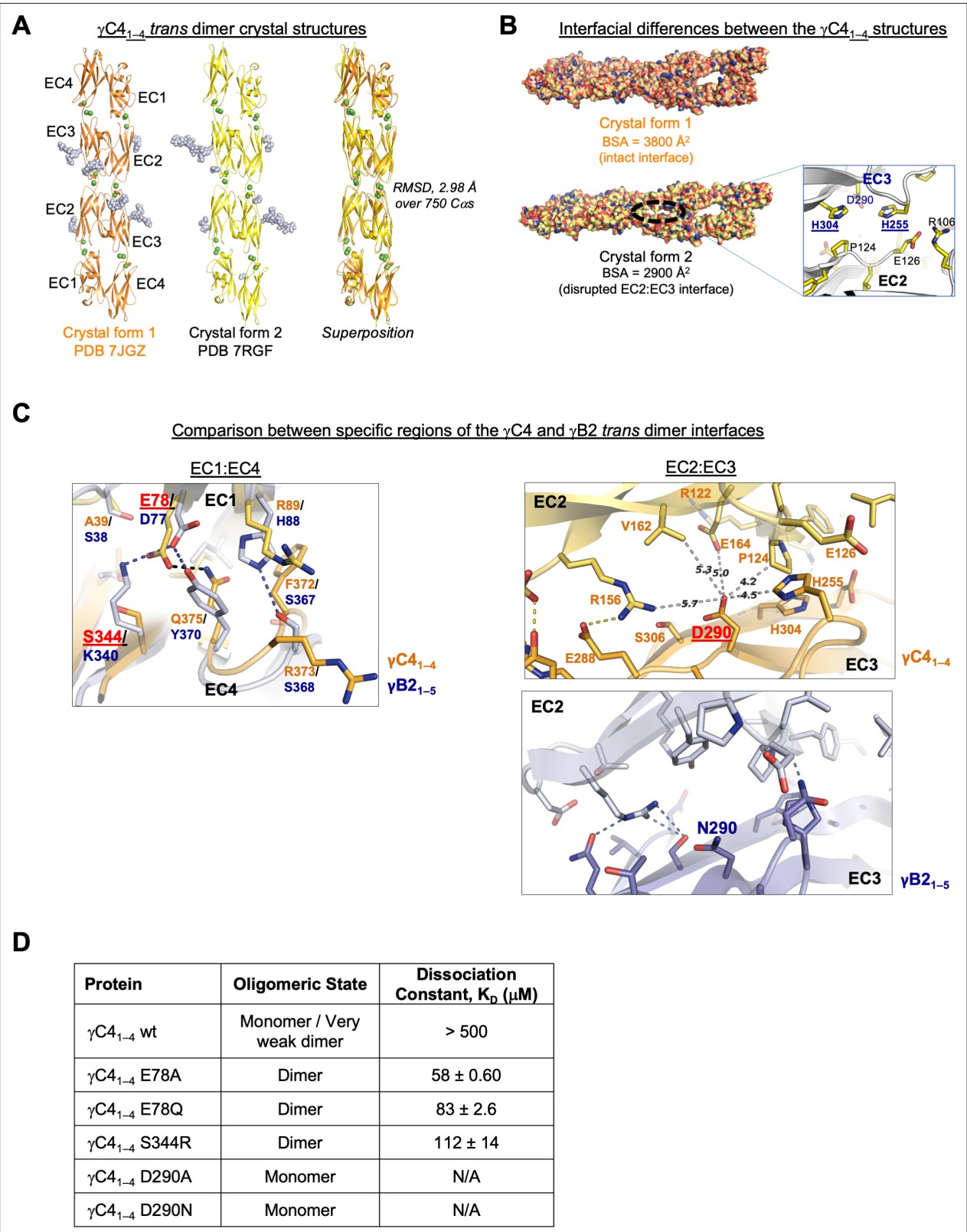

**Figure 3.** C-type clustered protocadherin (cPcdh) γC4 adopts an EC1–4-mediated head-to-tail *trans* dimer like alternate cPcdhs with a comparatively weak dimer affinity. (**A**) Ribbon diagrams of the γC4_EC1–4 *trans* dimer crystal structures obtained from two different crystal forms. Bound calcium ions are shown as green spheres and glycans are shown in pale blue spheres. (**B**) The two crystal structures have a markedly different *trans* interface buried surface area (BSA). *Left*, surface views of the two *trans* dimer crystal structures highlight the difference, with a gap apparent in the EC2:EC3 region of

*Figure 3 continued on next page*

*Figure 3 continued*

the interface in crystal form two that is absent from crystal form 1. Surfaces are colored by atom type with the carbons colored orange for crystal form one and yellow for crystal form 2. *Right*, close-up view of the gap region in the crystal form two dimer with the side chains depicted as sticks. The intact crystal form 1 γC4 dimer is similar overall to those of the published intact alternate α, β, γA, and γB Pcdhs and the published δ2 nonclustered (nc) Pcdh *trans* dimers (root mean square deviation [RMSD] over aligned Cαs 2.4–4.5 Å; *Figure 3—source data 2*). The published crystal structures of γA1, γA8, and γB3 also show partially disrupted *trans* interfaces though in differing regions of the interface (*Goodman et al., 2016a*, *Nicoludis et al., 2016*). (**C**) Comparison between the (i) EC1:EC4 and (ii) EC2:EC3 regions of the γC4 (orange) and γB2 (blue, PDB 5T9T) *trans* dimer interfaces. Potential hydrogen bonds are depicted as dashed black/yellow (γC4) or blue (γB2) lines. (i) Structural alignment of the EC1:EC4 portion of the γC4 and γB2 *trans* dimers highlights a possible destabilizing role for γC4 residue E78 since unlike its counterpart in γB2 (D77), it is not juxtaposed with a basic residue. (ii) Similarly, an additional negatively charged residue (D290) which occupies a central position in the γC4 EC2:EC3 interface may also contribute to γC4's comparatively weak *trans* dimer interaction. Distances between the D290 side chain and its nearest contacts are shown as dashed gray lines with distances given in Angstroms. (**D**) Sedimentation equilibrium analytical ultracentrifugation (AUC) experiments were conducted on γC4 EC1–4 wild-type (wt) and interface mutants to assess whether E78 and D290 negatively impact *trans* dimerization. Table details the oligomeric state and dissociation constants for each protein tested.

The online version of this article includes the following source data and figure supplement(s) for figure 3:

**Source data 1.** X-ray crystallography data collection and refinement statistics.

**Source data 2.** Overall structural similarity between cPcdh γC4, alternate cPcdhs, and non-clustered Pcdhs trans dimer structures.

**Figure supplement 1.** γC4 *trans* dimer crystal structures and *trans* interface analysis.

dimer interfaces (*Goodman et al., 2016a*; *Goodman et al., 2016c*; *Nicoludis et al., 2016*; *Figure 3B*, *Figure 3—figure supplement 1B*). However, the 2.4 Å structure had an apparently partially disrupted EC2:EC3 interface resulting in a total buried surface area of just 2900 Å$^2$ (*Figure 3B*). The difference between the two structures may be due to differences in the pH of the crystallization and its effect on the ionization state of the three histidines present in the EC2:EC3 interface (*Figure 3B*). The differences could also reflect distinct states of a dynamic interaction, as has previously been observed crystallographically (*Nicoludis et al., 2016*; *Goodman et al., 2016a*) and explored computationally for other cPcdh *trans* interactions (*Nicoludis et al., 2019*).

Despite the γC4 *trans* dimer sharing structural similarity and the interface having similar buried surface area as alternate α, β, γA, and γB cPcdhs and δ2 nonclustered Pcdhs (*Figure 3—source data 2*; *Cooper et al., 2016*; *Goodman et al., 2016a*; *Goodman et al., 2016c*; *Harrison et al., 2020*; *Hudson et al., 2021*; *Nicoludis et al., 2016*), its binding affinity is very weak. The two most structurally similar molecules to γC4 over their *trans*-interacting domains are cPcdh γB2 and nonclustered Pcdh19. γB2 and Pcdh19 have *trans* dimer $K_D$s of 21.8 and 0.48 µM, respectively (*Harrison et al., 2020*), while that of γC4 is >500 µM. Comparison between the γB2 and γC4 dimer interfaces highlighted two buried charges in the γC4 *trans* interface, E78 and D290, which could potentially contribute to the low interaction affinity (*Figure 3C*). To test this, we mutated these two residues to neutral amino acids and used AUC to determine whether the binding affinity increased: The two D290 mutations we tested, D290A and D290N, had no measurable impact on binding; but mutating E78 significantly increased the binding affinity with γC4$_{EC1-4}$ E78A showing a $K_D$ of 58 µM and γC4$_{EC1-4}$ E78Q, 83 µM (*Figure 3D*, *Figure 3—figure supplement 1C*). The equivalent residue to E78 in γB2 is also charged (D77) and forms a salt bridge with K340 in the γB2 dimer (*Figure 3C*). To assess whether generating a similar salt bridge in γC4 would compensate for the negative impact of E78 on dimer affinity we generated an S344R mutant. Similar to the E78 mutants, γC4$_{EC1-4}$ S344R also had a stronger binding affinity than wild-type with a $K_D$ of 112 µM (*Figure 3D*, *Figure 3—figure supplement 1C*). It appears then that E78 plays an important role in weakening cPcdh γC4's *trans* interaction although the functional reasons for γC4's weak *trans* interaction are unclear.

## cPcdh *cis* interactions are promiscuous with a range of interaction strengths

To systematically investigate cPcdh *cis* interactions, we coupled *cis*-interacting fragments of mouse β9, γA4, γA9, γB2, αC2, γC3, and γC5 to SPR chip surfaces. *Cis*-interacting fragments of three members from each of the β, γA, and γB subfamilies (β1, β6, β9, γA3, γA4, γA9, γB2, γB5, and γB7) alongside αC2, γC3, and γC5 fragments were flowed over the seven surfaces to detect their heterophilic binding (*Figure 4A*). Alternate α-cPcdhs, and the C-types αC1 and γC4 were not included in this study since EC6-containing fragments of these molecules cannot be expressed, although an α7$_{EC1-5}$/

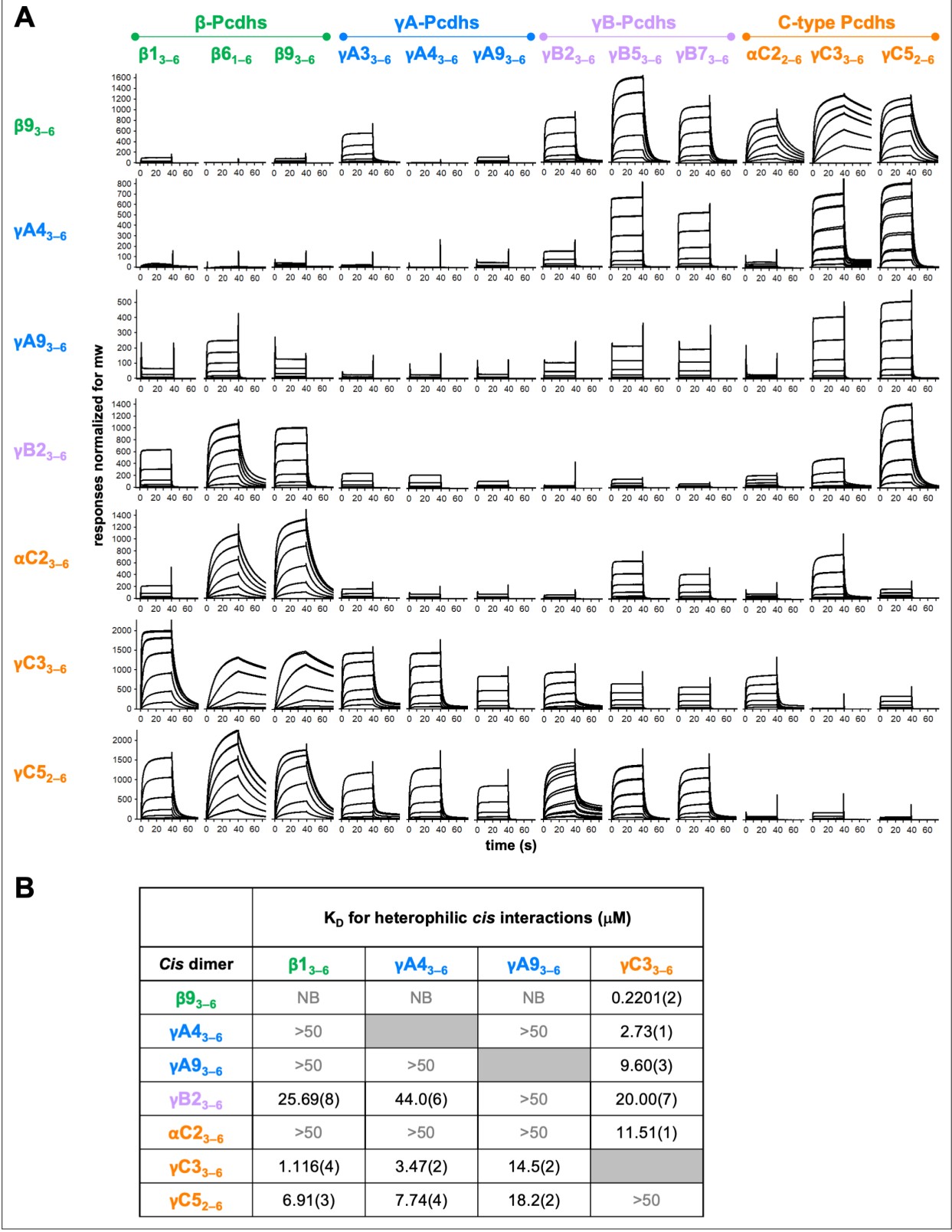

**Figure 4.** Clustered protocadherin (cPcdh) *cis* interactions are promiscuous with a preference for interfamily heterodimers. (**A**) Surface plasmon resonance (SPR)-binding profiles of cPcdh *cis* fragment analytes from all cPcdh subfamilies except alphas (shown in columns) flowed over individual surfaces coated with cPcdh *cis* fragments. Binding profiles for each surface are individually scaled and responses are normalized for molecular weight. (**B**) Table of dissociation constants calculated from the SPR data for the four monomeric analytes. The number in brackets represents the error of the fit

*Figure 4 continued on next page*

*Figure 4 continued*

based on analysis of duplicate responses. Binding signals were not detected for interactions labeled NB (no binding) while >50 represents interactions with $K_D$s > 50 μM, where an accurate $K_D$ cannot be determined.

The online version of this article includes the following source data and figure supplement(s) for figure 4:

**Source data 1.** Sedimentation equilibrium analytical ultracentrifugation data for cis SPR reagents.

**Figure supplement 1.** Calculation of *cis* interaction dissociation constants and the impact of an α-Pcdh EC5 on family-wide *cis* interactions.

**Figure supplement 2.** Range of clustered protocadherin (cPcdh) *cis* and *trans* dissociation constants, $K_D$s.

**Figure supplement 3.** Amino acid sequence alignment reveals conservation of *cis* interfacial residues within the alternate clustered protocadherin (cPcdh) subfamilies.

γC3$_{EC6}$ chimera was included among the analytes to assess the role of α7 EC5 (*Figure 4—figure supplement 1C*). Each of the analytes was also analyzed by AUC to determine their homophilic *cis*-interaction behavior (*Figure 4—source data 1*): Four analytes, β1$_{3–6}$, γA4$_{3–6}$, γA9$_{3–6}$, and γC3$_{3–6}$, are monomeric in solution as measured by AUC, therefore their SPR binding profiles could be analyzed to determine their heterophilic binding affinities (*Figure 4B*, *Figure 4—figure supplement 1A, B*). For the remaining analytes, due to the added complexity of their homophilic *cis* interactions in solution competing with their binding to the immobilized molecules, the SPR responses could not be analyzed to determine accurate $K_D$s (*Rich and Myszka, 2007*).

The data clearly demonstrate a wide range of *cis* dimerization affinities with strong heterophilic binding signals (500–2000 RU), with much weaker homophilic binding responses typically between 100 and 140 RU. The strongest heterophilic *cis* interactions are in the submicromolar range; for example, γC3/β9 can heterophilically *cis*-dimerize with a $K_D$ of 0.22 μM, while β9$_{3–6}$, γB2$_{3–6}$, αC2$_{2–6}$, and γC5$_{2–6}$ homodimerize with AUC-determined $K_D$s of 9–80 μM. In addition to uniformly weak homophilic interactions, within-subfamily *cis* interactions were consistently among the weakest observed although a number of intersubfamily interactions were also relatively weak (*Figure 4A*). For example, for the β9 surface comparatively weak binding was observed for all tested β and γA isoforms except γA3, with the monomeric β1, γA4, and γA9 producing low responses that could not be fit to a binding isotherm to calculate accurate $K_D$s (*Figure 4B*, *Figure 4—figure supplement 1B*). In contrast, robust binding to the β9 surface was observed for all γB and C-type isoforms. These data are consistent with the binding responses when β9 was used as an analyte over the other six surfaces, with weak to no binding observed over the γA4 and γA9 surfaces and robust responses over the γB2, αC2, γC3, and γC5 surfaces (*Figure 4A*). The γA4 and γA9 surfaces showed a similar pattern of binding behaviors, with weak to no binding observed for the γA and αC2 analytes, and robust binding for the γC-cPcdhs with $K_D$s for γC3$_{3–6}$ of 2.73 and 9.60 μM, respectively, over each surface (*Figure 4*, *Figure 4—figure supplement 1B*).

Overall, these SPR data show that cPcdh *cis* binding is generally promiscuous, with measurable *cis* interactions observed for 86% of pairs tested (using a 40 RU threshold). However, the wide range of binding responses and homo- and heterodimeric $K_D$s that span 0.2201 μM to no measurable interaction in solution suggests certain *cis* dimers will form preferentially to others. For the heterophilic binding pairs for which $K_D$s could be determined (*Figure 4B*, *Figure 4—figure supplement 1*, *Figure 4—figure supplement 2*), the alternate cPcdhs in particular, form markedly stronger *cis* heterodimers with members of different subfamilies, particularly γC3 and/or γC5, compared to their homodimeric and within-subfamily interactions. γC3 also formed stronger heterodimers with αC2 than with itself or γC5. Of note, αC2 and γC5 both form strong *cis* homodimers with $K_D$s of 8.9 and 18.4 μM, respectively, as determined from AUC experiments (*Figure 4—source data 1*), a magnitude similar to many of their heterodimeric interactions of 11.5 and 6.9–18.2 μM, respectively (*Figure 4B*).

In the next section, we rationalize *cis*-binding preferences in terms of the structural properties of *cis* dimers.

## The asymmetric *cis* dimer interface and *cis*-binding specificity

The crystal structure of the γB7 *cis* dimer revealed an asymmetric interaction, with the dimer formed by one protomer engaging using surface of both EC5 and EC6 and one protomer engaging using only EC6 (*Goodman et al., 2017*) with regions of EC6 overlapping in both EC5–6 and the EC6-only interfaces for all cPcdh subfamilies (*Thu et al., 2014*; *Goodman et al., 2017*). The asymmetric nature

of the *cis* interaction implies that for each dimer interaction there are two possible arrangements: one with protomer '1' forming the EC5–6 side and protomer '2' forming the EC6-only side and the second where protomer '1' forms the EC6-only side and '2' the EC5–6 side. These two configurations are distinct with different residue:residue interactions. Alternate α-Pcdhs, which can only form the EC5–6 side of the *cis* dimer, require coexpression with a 'carrier' cPcdh from another cPcdh subfamily, which can form the EC6-only side of the *cis* dimer, for robust delivery to the cell surface (*Thu et al., 2014*; *Goodman et al., 2017*). Although α-cPcdhs and γC4, which also requires a carrier for delivery to the cell surface, are likely to be extreme cases, sequence analysis alongside the low homodimerization ability of many cPcdh isoforms suggests many cPcdhs will more readily form one side of the *cis* interface than the other (*Goodman et al., 2017*).

We previously suggested that γA-cPcdhs will prefer to form the EC6-only side of the interface since they have a poorly conserved EC5 interface and do not form strong homodimers in solution (*Figure 4—source data 1*; *Goodman et al., 2017*). The C-type cPcdh γC3 also does not form *cis* homodimers in solution. However, as shown in *Figure 4*, γA-cPcdhs form strong heterodimers with γC3 with dissociation constants in the low-micromolar range (*Figure 4B* and *Figure 4—figure supplement 1B*). Structure-guided sequence analysis for the γA4/γC3 dimer in both EC6-only and EC5–6 possible orientations, using the available crystal structures of the γB7$_{EC3–6}$ *cis* dimer and monomeric γA4$_{EC3–6}$ (*Figure 5A* and *Figure 5—figure supplement 1*), suggests that γC3 prefers to form the EC5–6 side: γC3 has a number of residue differences in interface residues that are conserved among β, γA, and γB cPcdhs (V/L555, R/K558, W/V562, and S/R595) that seem likely to disfavor the EC6-only side of the interface and favor the EC5–6 side (*Figure 5—figure supplement 1B, C,*). Two of these residues, V555 and S595, result in a potential loss of EC6-only interface buried surface area and are shared with α-cPcdhs, which cannot occupy the EC6-only position (*Goodman et al., 2017*). Structural analysis further suggests that γC3-specific residue R558 would not be well accommodated from the EC6-only side, potentially causing van der Waals clashes (*Figure 5—figure supplement 1C*). By contrast, from the EC5–6 side R558 is positioned to form an additional salt bridge with γA4 residue E544 and a hydrogen bond with Y532, promoting dimer formation (*Figure 5A*; *Figure 5—figure supplement 1B*). γA4 residue E544 is positioned to form this salt bridge due to the EC6 A/A' loop region adopting a different arrangement in the γA4 crystal structure to that observed for γB2 and γB7 in their respective crystal structures (*Goodman et al., 2016a*; *Goodman et al., 2017*).

Based on our analysis, we generated mutants of both γA4 and γC3 targeting the EC6-only side of the interface and used size exclusion-coupled multiangle light scattering (SEC-MALS) to assess their preferred orientation on γA4/γC3 heterodimerization. In SEC-MALS wild-type γA4$_{EC3–6}$ and γC3$_{EC3–6}$ behave as monomers when run alone, and form a dimer when mixed in equimolar amounts (*Figure 5B*; *Figure 5—figure supplement 2A*). The V560R mutation (γB7 numbering, see methods for sequence alignment) is based on EC6-only impaired α-cPcdhs, and has been previously shown to block γB6's homophilic *cis* interaction in solution (*Goodman et al., 2017*). γA4 V560R did not dimerize with wild-type γC3, whereas γC3 V560R could still dimerize with wild-type γA4 (*Figure 5B*). Therefore, impairing γA4's EC6-only interface blocks γA4/γC3 dimer formation while impairing γC3's EC6-only interface does not (although the dimerization appears to be weaker compared to the wild-type γA4/γC3 *cis*-interacting pairs). We also generated a γC3-like mutant of γA4, K558R, which also targets the EC6-only interface. Like γA4 V560R, γA4 K558R also did not dimerize with wild-type γC3 in MALS and, when replicated, in SPR experiments (*Figure 5B*, *Figure 5—figure supplement 2B*). The reverse mutation in γC3, R558K, inhibited dimerization with wild-type γA4 (*Figure 5B*). Therefore, like the α-specific R560 residue, γC3-specific R558 has distinct effects on dimerization when in γA4 or γC3, inhibiting heterodimerization when mutated into γA4 but promoting heterodimerization in γC3. Together these data suggest that the γA4/γC3 dimer has a preferred orientation, with γA4 predominantly occupying the EC6-only position and γC3 the EC5–6 side. Our data also account for the fact that neither isoform homodimerizes in solution since the EC5–6 side would be impaired in the γA4 homodimer while the EC6 side would be impaired in the γC3 homodimer.

Next, we sought to test whether γA4 and γC3 preferentially adopt these specific positions in *cis* interactions with a γB isoform. To accomplish this we generated mutants of γB7 individually targeting the EC6-only interaction surface, γB7 Y532G, and the EC5–6 side, γB7 A570R, respectively (*Goodman et al., 2017*; *Figure 4—source data 1*). In SPR, γB7 Y532G had only a small impact on γA4 binding, while γB7 A570R abolished γA4 binding (*Figure 5C*). In contrast, γB7 Y532G prevented γC3 binding

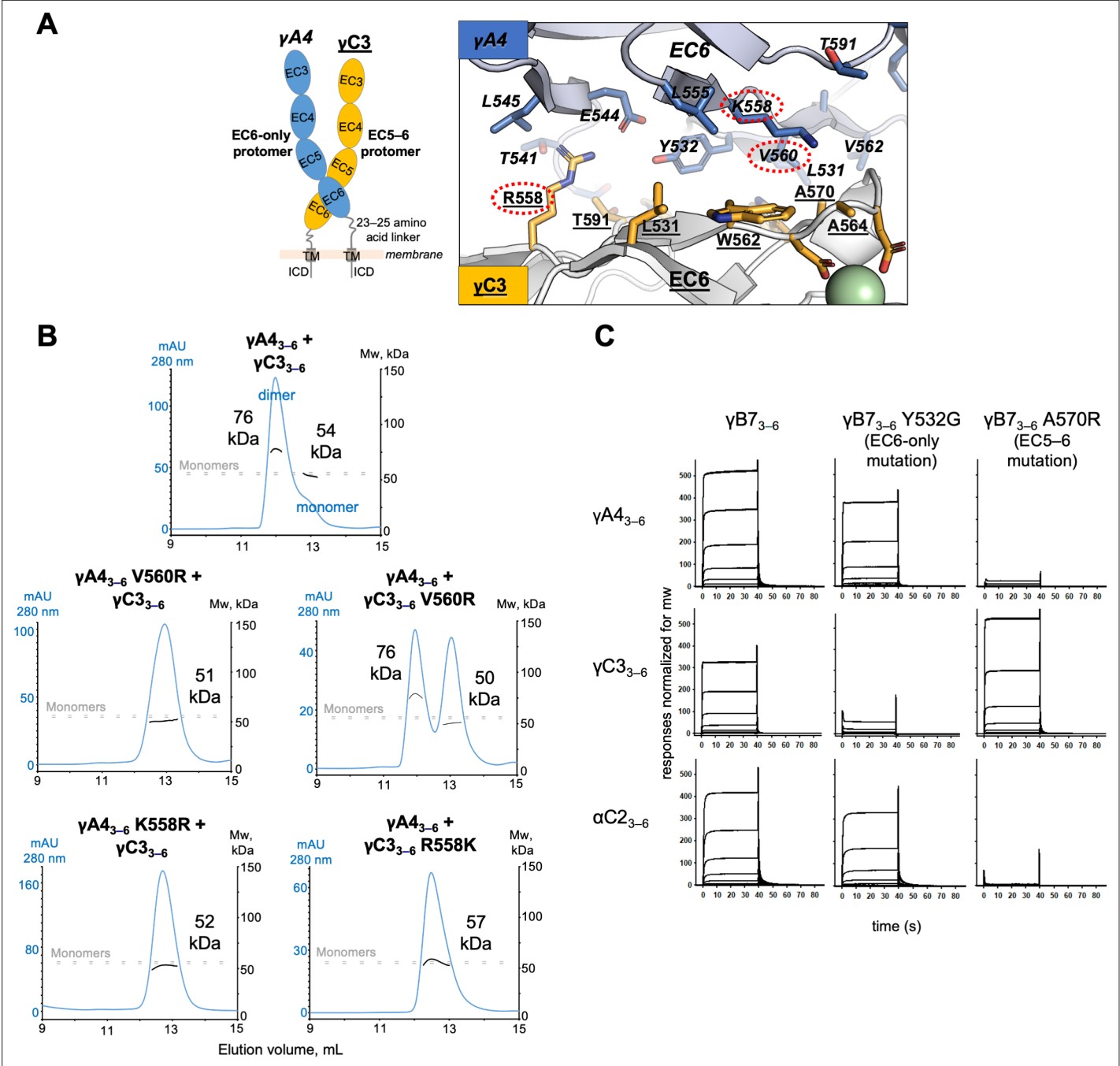

**Figure 5.** γA4 preferentially forms the EC6-only side and γC3 the EC5–6 side in *cis* dimers. (**A**) Structural model of γA4/γC3 *cis* dimer based on γB7_EC3–6 *cis* dimer and γA4_EC3–6 crystal structures (PDBs: 5V5X and 5SZQ). γA4 is shown adopting the EC6-only side (blue protomer) and γC3 is shown adopting the EC5–6 side (yellow protomer). *Left*, schematic of the γA4/γC3 EC3–6 *cis* dimer. *Right*, close-up view of the EC6:EC6 interface from the modeled *cis* dimer showing interfacial residue side chains. Bound calcium ions are shown as green spheres. Residues which were mutated in the panel B are circled in red. γB7 crystal structure numbering is used for both γA4 and γC3 residues. See methods for γA4 and γC3 alignment. Please note the model shown here is solely for hypothesis generation, since it is unlikely to be completely accurate. See methods for further details of structural modeling. (**B**) *Top*, size exclusion-coupled multiangle light scattering (SEC-MALS) data for an equimolar mixture of wild-type γA4_EC3–6 and γC3_EC3–6 showing dimer formation. Plot shows size exclusion absorbance at 280 nm trace (left axis), molecular weight of the eluant peaks (right axis), and the monomer molecular weights of γA4_EC3–6 and γC3_EC3–6 measured by mass spectrometry – 54.5 and 56.5 kDa, respectively – as dashed gray lines. Average molecular weight of the molecules in the dimer and monomer eluant peaks are labeled. *Middle*, SEC-MALS data for V560R mutants, which target the EC6-only side of the interface. *Bottom*, SEC-MALS data for residue 558 mutants. The γC3-like K558R mutation in γA4 inhibits heterodimer formation with wild-type γC3.

*Figure 5 continued*

Similarly, the γA4-like R558K in γC3 inhibits dimerization with wild-type γA4. (**C**) SPR-binding profiles for γB7$_{EC3-6}$ wild-type and *cis* interface mutants flowed over three individual wild-type *cis* fragment surfaces. The two mutations specifically target one side of the *cis* interface.

The online version of this article includes the following figure supplement(s) for figure 5:

**Figure supplement 1.** Structure-guided sequence analysis of γA4 and γC3 *cis* interactions.

**Figure supplement 2.** γA4 and γC3 *cis* fragments behave as monomers in size exclusion-coupled multiangle light scattering (SEC-MALS) and mutating γA4 to make it more like γC3 prevents γA4/γC3 *cis* heterodimerization.

while γB7 A570R showed robust γC3 binding (*Figure 5C*). These results suggest that γA4/γB7 and γC3/γB7 *cis* heterodimers also have preferred orientations with γA4 and γC3 maintaining their preferences for the EC6-only and EC5–6 positions, respectively. Additionally, SPR data for the γB7 mutants over the αC2 surface suggest αC2 preferentially occupies the EC6-only side in αC2/γB7 dimers (*Figure 5C*). This is notable since αC2 forms robust *cis* homodimers and therefore, like γB7, can presumably readily occupy both positions in its homophilic interactions, implying that the αC2/γB7 orientation preference could be specific to the particular heterodimer pairing. However, since this interpretation is based on a single mutation further interrogation of αC2's interactions would be required to be conclusive. A broader examination of orientation preferences among *cis* dimer pairings beyond those of molecules with weak *cis* homodimer affinities, such as γA4 and γC3 examined here, could be instructive.

## Discussion

### *Trans* specificity

The results of this study add to our current understanding of cPcdhs in a number of ways. First, they reveal a remarkable level of specificity in *trans* homophilic interactions since in no case was a heterophilic *trans* interaction detected in our SPR measurements. Prior data have clearly indicated that cPcdhs exhibit a preference for homophilic *trans* interactions but the extent of this specificity was not established in quantitative terms but were, rather, based on cell aggregation experiments. The SPR experiments with cPcdhs reported here show no evidence of cross-interaction between nonidentical cPcdh isoforms. This level of specificity is unusual for cell–cell recognition proteins, as significant intrafamily interactions are evident in most other families examined to date including type I cadherins (*Katsamba et al., 2009*; *Vendome et al., 2014*), type II cadherins (*Brasch et al., 2018*), DIPs and Dprs (*Cosmanescu et al., 2018*), sidekicks (*Goodman et al., 2016b*), and nectins (*Harrison et al., 2012*). Even the nonclustered δ- protocadherins, which are preferentially homophilic and utilize an antiparallel EC1–4 interface like the cPcdhs (*Cooper et al., 2016*; *Harrison et al., 2020*; *Modak and Sotomayor, 2019*), show heterophilic intra-family *trans* interactions, though they show no cross-reactivity with cPcdhs (*Harrison et al., 2020*).

High-fidelity homophilic interaction is a strict requirement of the chain-termination model for the barcoding of vertebrate neurons and has been accomplished through the exploitation of a multi-domain interface of almost 4000 Å$^2$ (*Nicoludis et al., 2019*) that enables the positioning of enough 'negative constraints' (*Sergeeva et al., 2020*) to preclude the dimerization of about 1600 heterophilic pairs of 58 mouse cPcdh isoforms (*Rubinstein et al., 2017*). Dscams accomplish the same task for thousands of isoforms by exploiting the combinatorics made possible by a three-domain interface where each domain interacts largely independently with an identical domain on its interacting partner (see discussion in *Zipursky and Grueber, 2013*). Although it is likely that Dscams dimerize with a comparable level of homophilic specificity to that of cPcdhs, the evidence is based on a semiquantitative ELISA-type assay of recombinant multimerized isoforms (*Wojtowicz et al., 2007*) and AUC experiments on a few select isoforms (*Wu et al., 2012*).

### *Cis* interactions

Despite early evidence that *cis* interactions are promiscuous, the data reported here indicate that this generalization needs to be significantly refined. Functional mutagenesis studies have already established that alternate α cPcdhs and the C-type γC4 do not form intrasubtype *cis* interactions and can only reach the cell surface when mediated by heterophilic *cis* interactions with members of

other subtype families (*Goodman et al., 2017*; *Thu et al., 2014*). The data presented in *Figure 4* indicate that this is an extreme example of quite general behavior: intrasubtype *cis* interactions are invariably weaker than intersubtype interactions. However, unlike α cPcdhs, most cPcdhs can reach the cell surface on their own. This includes β1, all γA-Pcdhs, and γC3 which do not form measurable homodimeric *cis* interactions in our solution-based AUC experiments. We have attributed this to their presence on the restricted 2D surface of membranes which can promote *cis* dimerization (*Wu et al., 2013*) whereas biophysical experiments are carried out in a 3D solution environment (*Goodman et al., 2016a*). (There may of course be other, still undetermined, factors involved in cPcdh cell surface transport [*Phillips et al., 2017*].) Therefore, although our biophysical experiments demonstrate that intrasubtype *cis* interactions are comparatively weak and, in some cases undetectable in solution, intrasubtype *cis* dimers likely assemble when constrained in more native membrane environments. As such, while α cPcdhs and γC4 are obligate participants in *cis* heterodimers, at least in their cell surface transport, our data show that the remaining cPcdhs are preferentially, although not exclusively, participants in *cis* heterodimers.

The *cis*-binding preferences indicated by our data can be largely understood in terms of the asymmetric interface discussed above. Specifically, different isoforms preferentially form one side of the *cis* dimer: for example, the EC6-only side for cPcdh-γA4 and the EC5–6 side for cPcdh-γC3. Homodimerization requires participation of single isoform on both sides of an interface posing challenges in the optimization of binding affinities since, in some cases, the same residue must participate in different intermolecular interactions. Given significant sequence conservation in all members of an alternate cPcdh subfamily (*Figure 4—figure supplement 3*) even intrasubfamily heterophilic interactions are more difficult to optimize relative to intersubfamily heterodimerization where there are no constraints on the two interacting surfaces. Additionally, the robust cell surface delivery of many cPcdhs in cells expressing only a single isoform also suggests that all carrier isoforms – β-, γA-, and γB-cPcdhs, plus C-types αC2, γC3, and γC5 – can fill both the EC6 and EC5–6 roles, as *cis* dimer formation is thought to be required for cell surface export (*Goodman et al., 2017*; *Goodman et al., 2016a*; ; *Thu et al., 2014*). Therefore side preferences are most likely not absolute for carrier cPcdh isoforms and may vary among individual isoform and/or subtype pairings.

## Functional implications of cPcdh interactions

The functional role of precise *trans* homophilic specificity in ensuring high-fidelity discrimination between neuron self and nonself has been discussed previously (*Rubinstein et al., 2017*; *Rubinstein et al., 2015*) and is summarized above. It is an essential feature of the chain-termination model. The role of promiscuous *cis* interactions can also be understood in terms of this model in that *cis* promiscuity enables the formation of a large and diverse set of *cis* dimers that can only form long molecular zippers when all isoforms are matched. However, the results of this study reveal strong preferences for intersubgroup heterophilic interactions whose biological rationale is uncertain. cPcdhs from the three subfamilies have been shown to act cooperatively in certain neuronal contexts although whether this relates to their *cis* interactions is unknown (*Hasegawa et al., 2016*; *Ing-Esteves et al., 2018*).

One possible advantage of weak homophilic *cis* interactions would be to ensure that once reaching the cell surface a diverse set of *cis* dimers forms. This explanation implicitly assumes that most isoforms (except for α-Pcdhs and γC4) reach the surface as homodimers that must then quickly dissociate and form more stable heterodimers. Another explanation posits that homotypic zippers consisting solely of *cis* homodimers are kinetically easier to form than heterotypic zippers since in a homotypic zipper, either 'wing' of the new *cis* dimer can form *trans* interactions with the wing at the chain terminus. In contrast, in a heterodimeric zipper, only one wing can form homophilic interactions with the chain terminus (*Figure 1D*). A preference for homotypic zippers would then reduce the diversity required in the chain-termination model since, in this model, it is essential that all isoforms be incorporated into a growing zipper. The formation of long homotypic zippers might lead to a repulsive phenotype even when mismatches are present.

However, these explanations would not fully account for interfamily heterophilic preferences. One possibility is suggested by the observation that C-types are often highly expressed compared to alternate cPcdhs, for example in Purkinje cells (*Esumi et al., 2005*; *Kaneko et al., 2006*). To ensure sufficient diversity in growing zippers, it would then be important to ensure that zippers that are formed are not overly enriched in C-type isoforms as would be accomplished through preferential heterophilic

*cis* interactions. This same logic would also pertain to alternate cPcdhs in cases where one subfamily is more heavily expressed than another.

C-type cPcdhs have different functions than alternate cPcdhs and these are reflected in different expression patterns. For example, αC2 can be alone responsible for tiling (*Chen et al., 2017*). (Of note, in the chain-termination model, a completely homophilic zipper is sufficient to initiate self-avoidance facilitating tiling.) On the other hand, γC4, which has a unique and crucial role in neuronal survival (*Garrett et al., 2019*), requires coexpression with another cPcdh isoform for robust cell surface expression and therefore is likely unable to act in isolation (*Thu et al., 2014*). Furthermore, as detailed above, γC4 has a much weaker *trans* interaction affinity than any other cPcdh isoform measured to date, although it is still able to mediate cell aggregation when delivered to the cell surface (*Thu et al., 2014*). The presence of E78 appears in large part to be responsible for this weak affinity. It is unclear whether γC4's weak *trans* affinity plays any functional role, although a weak homodimer interaction may facilitate extracellular interactions with other, currently unidentified, proteins. More generally, it seems likely that different intracellular interactions account for the specialized functions of C-type Pcdhs. The cytoplasmic domain plays an important role in the activation of Wnt, WAVE, and other signaling cascades (*Chen et al., 2009*; *Fukuda et al., 2008*; *Keeler et al., 2015*; *Mah and Weiner, 2017*; *Onouchi et al., 2015*; *Pancho et al., 2020*). In some cases, the cytoplasmic domains of a subset or even a single cPcdh isoform activates a specific signaling cascade. For example, cPcdh γC3 is the only isoform able to interact and inhibit Axin1, a Wnt pathway activator (*Mah et al., 2016*). Of note, γ-cPcdh intracellular domains consist of a C-terminal constant region common to all γ isoforms (including the three γ C-types) and a membrane-proximal variable region consisting of ~100 residues that could account for the unique intracellular interactions and signaling of individual isoforms. Additionally, it is possible that extracellular interactions to molecules from other families, such as Neuroligins, may account for some distinctions in function (*Molumby et al., 2017*; *Steffen et al., 2021*).

Overall, the results of this study demonstrate the remarkable tuning of the interactions among cPcdh family members: homophilic *trans* interactions are remarkably specific despite the high level of sequence identity among family members while *cis* interactions, though somewhat promiscuous, also appear designed to have binding preferences of still uncertain function. These binding properties match requirements of the 'isoform-mismatch chain-termination model' for neuronal self- vs nonself-discrimination in which all expressed cPcdh isoforms assemble into intercellular zippers formed by alternating promiscuous *cis* and matched *trans* interactions with assembly size dictated by the presence or absence of mismatched isoforms. It remains to be seen whether such assemblies can be observed in vivo and how they control downstream signaling pathways.

# Materials and methods
## Protein production and purification

cDNAs for mouse cPcdh ectodomain fragments, excluding the predicted signal sequences, were cloned into a pαSHP-H mammalian expression vector (a kind gift from Daniel J. Leahy, John Hopkins University) modified with the human binding immunoglobulin protein (BiP; MKLSLVAAMLLLLSAARA) signal sequence and a C-terminal octa-histidine tag (*Rubinstein et al., 2015*). The signal sequences were predicted using the SignalP 4.0 server (*Petersen et al., 2011*). Point mutations were introduced into cDNA constructs using the KOD hot start polymerase (Novagen) following the standard Quikchange protocol (Stratagene).

Suspension-adapted HEK293 Freestyle cells (Invitrogen) in serum-free media (Invitrogen) grown and maintained at 37°C and 10% carbon dioxide were used for protein expression. FreeStyle 293F cell line has been authenticated and verified negative for mycoplasma using PCR testing (Thermo Fisher). The plasmid constructs were transfected into cells using polyethyleneimine (Polysciences Inc) (*Baldi et al., 2012*). Media was supplemented with 10 mM $CaCl_2$ 4 hr after transfection. Conditioned media was harvested ~6 days after transfection and the secreted proteins were purified using batch nickel-nitrilotriacetic acid (Ni-NTA) affinity chromatography followed by size exclusion chromatography over Superdex 200 26/60 column (Cytiva) on an AKTA pure fast protein liquid chromatography system (Cytiva). Purified proteins were concentrated to >2 mg/ml in 10 mM Tris–Cl, pH 8.0, 150 mM NaCl, 3 mM $CaCl_2$, and 100–250 mM imidazole pH 8.0 and stored at 4°C for short-term use or flash frozen in liquid nitrogen for long-term storage at −80°C.

Constructs encoding biotinylated cPcdh fragments for immobilization in SPR experiments were prepared by insertion of an Avi-tag (GLNDIFEAQKIEWHE)-encoding sequence between the octa-histidine tag and stop codon. These were cotransfected with a plasmid encoding the biotin-Ligase BirA from *E. coli* (Lys2–Lys321) with a BiP signal sequence and a C-terminal endoplasmic reticulum-retention signal (DYKDEL) (*Barat and Wu, 2007*). The expression and BirA plasmids were mixed at a 9:1 ratio for transfection and 50 µM Biotin (Sigma) was added to the media 4 hr post-transfection. Purification was carried out exactly as for the nonbiotinylated constructs and biotinylation was confirmed by western blot using NeutrAvidin-HRP (Thermo Fisher).

## Sedimentation equilibrium AUC

| Protein | Imidazole pH 8.0 (mM) | Spin speeds (rpm) |
|---|---|---|
| α4 EC1–5 | 100 | 9000, 11,000, 13,000, 15,000 |
| α7 EC1–5 L301R | 100 | 9000, 11,000, 13,000, 15,000 |
| α12 EC1–5 *(poorly behaved)* | 200 | 11,000, 14,000, 17,000, 20,000 |
| γB4 EC1–5 | 200 | 11,000, 14,000, 17,000, 20,000 |
| γB5 EC1–4-AVI | 200 | 11,000, 14,000, 17,000, 20,000 |
| γC5 EC1–5 S116R | 200 | 11,000, 14,000, 17,000, 20,000 |
| β6 EC1–4 | 100 | 9000, 11,000, 13,000, 15,000 |
| β6 EC1–4-AVI tag | 200 | 11,000, 14,000, 17,000, 20,000 |
| β6 EC1–4 R41N | 200 | 11,000, 14,000, 17,000, 20,000 |
| β6 EC1–4 S117I | 200 | 11,000, 14,000, 17,000, 20,000 |
| β6 EC1–4 L125P | 200 | 11,000, 14,000, 17,000, 20,000 |
| β6 EC1–4 E369K | 200 | 11,000, 14,000, 17,000, 20,000 |
| β6 EC1–4 Y371F | 200 | 11,000, 14,000, 17,000, 20,000 |
| β6 EC1–4 R41N/S117I *(precipitates)* | 200 | 11,000, 14,000, 17,000, 20,000 |
| β6 EC1–4 R41N/E369K | 200 | 11,000, 14,000, 17,000, 20,000 |
| β6 EC1–4 S117I/L125P | 200 | 11,000, 14,000, 17,000, 20,000 |
| β6 EC1–4 R41N/S117I/L125P | 200 | 11,000, 14,000, 17,000, 20,000 |
| β6 EC1–4 R41N/S117I/E369K | 200 | 11,000, 14,000, 17,000, 20,000 |
| β6 EC1–4 R41N/E369K/Y371F | 200 | 11,000, 14,000, 17,000, 20,000 |
| β6 EC1–4 R41N/S117I/L125P/ E369K/Y371F | 200 | 11,000, 14,000, 17,000, 20,000 |
| β1 EC3–6 | 200 | 12,000, 16,000, 20,000, 24,000 |
| β6 EC1–6 | 250 | 9000, 11,000, 13,000, 15,000 |
| β9 EC3–6 | 200 | 11,000, 14,000, 17,000, 20,000 |
| γA3 EC3–6 | 200 | 11,000, 14,000, 17,000, 20,000 |
| γA9 EC3–6 | 200 | 11,000, 14,000, 17,000, 20,000 |
| γB7 EC3–6 A570R | 200 | 13,000, 17,000, 21,000, 25,000 |
| αC2 EC3–6-AVI tag | 200 | 11,000, 14,000, 17,000, 20,000 |
| γC5 EC2–6 | 250 | 9000, 11,000, 13,000, 15,000 |
| γC4 EC1–4 | 250 | 11,000, 14,000, 17,000, 20,000 |
| γC4 EC1–4 D290A | 250 | 11,000, 14,000, 17,000, 20,000 |
| γC4 EC1–4 D290N | 250 | 11,000, 14,000, 17,000, 20,000 |

*Continued on next page*

*Continued*

| Protein | Imidazole pH 8.0 (mM) | Spin speeds (rpm) |
|---|---|---|
| γC4 EC1–4 E78A | 250 | 11,000, 14,000, 17,000, 20,000 |
| γC4 EC1–4 E78Q | 250 | 11,000, 14,000, 17,000, 20,000 |
| γC4 EC1–4 S344R | 250 | 11,000, 14,000, 17,000, 20,000 |

Experiments were performed in a Beckman XL-A/I analytical ultracentrifuge (Beckman-Coulter, Palo Alto CA, USA), utilizing six-cell centerpieces with straight walls, 12-mm path length and sapphire windows. Protein samples were dialyzed overnight and then diluted in 10 mM Tris–Cl, pH 8.0, 150 mM NaCl, 3 mM $CaCl_2$ with 100–250 mM imidazole pH 8.0, as detailed in the above table. The samples were diluted to an absorbance of 0.65, 0.43, and 0.23 at 10 and 280 nm in channels A, B, and C, respectively. For each sample, buffer was used as blank. The samples were run in duplicate at four speeds as detailed in the above table. The lowest speed was held for 20 hr then four scans were conducted with 1-hr interval, the subsequent three speeds were each held for 10 hr followed by four scans with 1-hr interval each. Measurements were taken at 25°C, and detection was by UV at 280 nm or interference. Solvent density and protein v-bar at both temperatures were determined using the program SednTerp (Alliance Protein Laboratories, Corte Cancion, Thousand Oaks, CA, USA). The molecular weight of each protomer used in AUC experiments was determined by MALDI mass spectrometry. For the calculation of dimeric $K_D$ and apparent molecular weight, all data were used in a global fit, using the program HeteroAnalysis (http://www.biotech.uconn.edu/auf). The calculation of the tetramer $K_D$s was done with the program Sedphat (http://www.analyticalultracentrifugation.com/sedphat/index.htm).

## SPR binding experiments

SPR-binding experiments were performed using a Biacore T100 biosensor equipped with a Series S CM4 sensor chip, immobilized with NeutrAvidin over all four flow cells. NeutrAvidin immobilization was performed in HBS-P (HEPES-Buffered Saline-P20) buffer, pH 7.4 at 32°C, over all four surfaces using amine-coupling chemistry as described in *Katsamba et al., 2009*, resulting in approximately 10,000 RU of NeutrAvidin immobilized (*Katsamba et al., 2009*). Binding experiments were performed at 25°C in a running buffer containing 10 mM Tris–Cl, pH 8.0, 150 mM NaCl, 3 mM $CaCl_2$, 20 mM imidazole, 0.25 mg/ml BSA (Bovine Serum Albumin), and 0.005% (vol/vol) Tween-20 unless otherwise noted.

C-terminal biotinylated fragments were tethered over individual NeutrAvidin-immobilized flow cells (shown in the left column of each *Figures 2, 4, and 5C*, *Figure 2—figure supplement 1*, *Figure 2—figure supplement 2B*, *Figure 4—figure supplement 1*, and *Figure 5—figure supplement 2B*) at 2300–3000 RU, depending on the experiment, using a flow rate of 20 µl/min. A NeutrAvidin-immobilized flow cell was used as a reference in each experiment to subtract bulk refractive index changes. The analytes tested in each experiment are listed at the top row. All analytes (with exceptions for the *cis*-interacting pairs γC3$_{3-6}$/β9$_{3-6}$, in both orientations, and β6$_{1-6}$/γC3$_{3-6}$ in *Figure 4A*, discussed below) were tested at six concentrations ranging between 24, 8, 2.667, 0.889, 0.296, and 0.099 µM, prepared using a threefold dilution series. γC3$_{3-6}$ binding over β9$_{3-6}$ (*Figure 4A*) was tested at five concentrations from 8 to 0.099 µM.

For all experiments, analyte samples were injected over the captured surfaces at 50 µl/min for 40 s, followed by 180 s of dissociation phase, a running buffer wash step and a buffer injection at 100 µl/min for 60 s. Protein samples were tested in order of increasing concentration, and within the same experiment the entire concentration series was repeated to confirm reproducibility. Every three binding cycles, buffer was used as an analyte instead of a protein sample to double reference the binding responses by removing systematic noise and instrument drift. The resulting binding curves were normalized for molecular weight differences according to data provided by mass spec for each molecule. The data were processed using Scrubber 2.0 (BioLogic Software). To provide an estimate of the number of possible heterophilic binding pairs, we have used a cutoff of 40 RU, which is the lowest signal that can be observed for a homodimeric *cis* fragment pair, γB2$_{3-6}$.

In *Figure 4A*, β6$_{1-6}$ and β9$_{3-6}$ were tested over γC3$_{3-6}$ at six concentrations ranging from 900 to 3.7 nM, which is 27-fold lower than the other interactions, prepared using a threefold dilution series

in a running buffer containing increased concentrations of imidazole (100 mM) and BSA (0.5 mg/ml) to minimize nonspecific interactions. For these two interactions, although analyte samples were injected over the captured surfaces at 50 μl/min for 40 s, the dissociation phase was monitored for 300 s to provide additional time for complex dissociation. Nevertheless, higher analyte concentrations produced binding profiles that were not reproducible, most likely due to the fact that bound complexes could not dissociate completely at these higher concentrations.

For the calculation of heterophilic $K_D$s for the monomeric *cis* fragments $\beta1_{3–6}$, $\gamma A4_{3–6}$, $\gamma A9_{3–6}$, and $\gamma C3_{3–6}$ over each of the six surfaces, except $\beta9_{3–6}$, the duplicate binding responses were fit globally, using an 1:1 interaction model and a single $K_D$ was calculated as the analyte concentration that would yield 0.5 $R_{max}$ and a fitting error, indicated in brackets. $K_D$s lower than 24 μM were calculated using an independent $R_{max}$. For $K_D$s greater 24 μM, the $R_{max}$ was fixed to a global value determined by the $R_{max}$ of a different cPcdh analyte tested over the same surface during the same experiment that showed binding above 50% and therefore produced a more accurate $R_{max}$. For $K_D$s > 50 μM, a lower limit is listed since at the analyte concentrations used (0.098–24 μM), accurate $K_D$s could not be determined, even when the $R_{max}$ is fixed. NB (no binding) represents interactions that did not yield any binding signal. The binding curves of $\gamma C3_{3–6}$ over the $\beta9_{3–6}$ did not come to equilibrium during the time-course of the experiment, so a kinetic analysis was performed to calculate a $K_D$ (*Figure 4—figure supplement 1A*). Binding of $\gamma C3_{3–6}$ was tested using a concentration range of 900–0.411 nM prepared using a threefold dilution series in a running buffer containing increased concentrations or imidazole (100 mM) and BSA (0.5 mg/ml) to minimize any nonspecific interactions. Protein samples were injected over the captured surfaces at 50 μl/min for 90 s, followed by 420 s of dissociation phase, a running buffer wash step and a buffer injection at 100 μl/min for 60 s. Protein samples were tested in order of increasing concentration in triplicate to confirm reproducibility. Every three binding cycles, buffer was used as an analyte instead of a protein sample to double reference the binding responses by removing systematic noise and instrument drift. The binding data were analyzed using an 1:1 interaction model to calculate the kinetic parameters and the $K_D$.

## K562 cell aggregation assays

Full-length cPcdhs β6 and β8 cDNAs were cloned into the pMax expression vectors encoding C-terminal mCherry- or mVenus-tagged cPcdh proteins, then transfected into K562 cells (ATCC CCL243) as previously described (*Goodman et al., 2017*; *Thu et al., 2014*). K-562 bone marrow chronic myelogenous leukemia cell line has been authenticated and verified negative for mycoplasma using PCR testing (ATCC). Point mutants were generated using the QuikChange method (Stratagene). In brief, K562 cells were cultured at 37°C with 5% $CO_2$ in Dulbecco's modified Eagle medium with GlutaMAX (GIBCO) supplemented with 10% fetal bovine serum and 1% penicillin–streptomycin for 2 days. Next, cells were counted, centrifuged, and resuspended at a density of $1.5 \times 10^4$ cells/μl in SF Cell Line 4D-Nucleofector Solution SF with supplement according to the manufacturer's instructions (Lonza). 2 μg of each Pcdh expression construct were transfected into 20 μl of the K562 cell suspension by electroporation using an Amaxa 4D-Nucleofector (Lonza). Transfected cells were transferred to a 24-well plate in 500 μl of medium per well and incubated overnight at 37°C and 5% $CO_2$. Cells then were mixed, reincubated with gentle rocking for 4 hr, then imaged with an Olympus IX73 fluorescent microscope to determine the extent of aggregation.

## Size exclusion-coupled multiangle light scattering

SEC-MALS experiments were performed using a Superdex 200 Increase 3.2/300 size exclusion column on an AKTA FPLC system (Cytiva) coupled to inline static light scattering (Dawn Heleos II, Wyatt Technology), differential refractive index (Optilab rEX, Wyatt Technology), and UV detection. Purified cPcdh proteins were diluted to 18 μM in running buffer (150 mM NaCl, 10 mM Tris–Cl, pH 8, 3 mM $CaCl_2$, 200 mM imidazole, pH 8) and 50 or 100 μl samples were run at a flow rate of 0.5 ml/min at room temperature. Mixtures of cPcdh fragments were prepared in the same buffer at final concentrations of 18 μM for each protein and run under the same conditions. Data were analyzed using ASTRA software (Wyatt Technologies).

During SEC-MALS experiments, a dimer/monomer equilibrium is established as proteins move through the size exclusion chromatography column, which is influenced by the $K_D$ of the interaction. The concentrations used in the current experiments (18 μM for each cPcdh fragment), although above

the $K_D$ of 3 µM for the γC3/γA4 *cis* interaction, are not sufficiently high for all the *cis* fragments to be bound into heterodimers, leaving a significant population of molecules as monomers, resulting in apparent molecular weights of ~76 kDa for the dimeric species compared to the predicted molecular weight for a dimer of ~108 kDa.

## X-ray crystallography

Crystallization screening of γC4$_{1–4}$ using the vapor diffusion method yielded two protein crystal forms: The first crystal form crystals were grown using a protein concentration of 7 mg/ml in 10% (wt/vol) PEG8000, 20% ethylene glycol, 10% Morpheus Amino Acids (Molecular Dimensions), and 0.1 M Morpheus Buffer System 2 (Hepes/MOPS buffer; Molecular Dimensions), pH 7.5. No additional cryo-protection was required for this crystal form. The second crystal form crystals were grown using a protein concentration of 7 mg/ml in 1 M LiCl, 0.1 M Mes pH 6.0, and 10% (wt/vol) PEG 6000. The crystal used for data collection was cryoprotected in the crystallization condition plus 30% (wt/vol) glycerol. X-ray diffraction data for each crystal form were collected at 100 K from single crystals at Northeastern Collaborative Access Team (NE-CAT) beamline 24ID-E at the Advanced Photon Source, Argonne National Laboratory.

## γC4$_{1–4}$ crystal form 1: diffraction anisotropy and pseudosymmetry

The X-ray diffraction data for the first crystal form showed strong diffraction anisotropy, with relatively strong diffraction along *c** and much weaker diffraction along *a** and *b** (*Figure 3—figure supplement 1A*). These data were therefore truncated using ellipsoidal limits with using a 3.0 F/sigma cutoff along each of the three principal crystal axes as implemented in the UCLA Diffraction Anisotropy Server (*Strong et al., 2006*) to 4.6/3.9/3.5 Å. The completeness within the applied ellipsoidal resolution limits was 96.8% (*Figure 3—source data 1*).

## γC4$_{1–4}$ crystal form 1: crystal structure phasing and refinement

The γC4$_{1–4}$ crystal structure was solved by molecular replacement using Phaser (*McCoy et al., 2007*), implemented in CCP4 (*Winn et al., 2011*). The γC5$_{EC1–3}$ crystal structure (PDB: 4ZPO) modified using a sequence alignment to γC4 with Phenix's MRage program (*Liebschner et al., 2019*) was used as a search model. Following an initial round of rigid body refinement in Phenix (*Liebschner et al., 2019*) the EC domain 4 from the α7$_{EC1–5}$ crystal structure (PDB: 5DZV) was manually placed into the electron density map, using structural alignment to the EC1–3 regions as a guide. The resulting model was subjected to a further round of rigid body refinement. At this stage there was clear difference density for the interdomain calcium ions and covalently linked glycans not present in the models. Iterative model building using Coot (*Emsley et al., 2010*) and maximum-likelihood refinement using Phenix (*Liebschner et al., 2019*) was subsequently conducted. The higher resolution (2.4 Å) crystal form two crystal structure (see below) was used as a reference model in later rounds of iterative model building and refinement to guide the local geometry choices in this lower resolution structure. Final refinement statistics are given in *Figure 3—source data 1*.

## γC4$_{1–4}$ crystal form 2: data processing, phasing, and refinement

The γC4$_{1–4}$ crystal form two dataset was indexed using XDS (*Kabsch, 2010*) and scaled using AIMLESS (*Evans and Murshudov, 2013*). The data were spherically truncated with high resolution limit of 2.4 Å. Data collection statistics are given in *Figure 3—source data 1*.

The γC4$_{1–4}$ crystal form two crystal structure has two molecules in the asymmetric unit was solved by molecular replacement using Phaser (*McCoy et al., 2007*), implemented in Phenix (*Liebschner et al., 2019*), using the EC2–3 portion of the *trans* dimer from the γC4$_{1–4}$ crystal form one crystal structure early in refinement as a search model. The molecular replacement solution was then subjected to an initial round of rigid body refinement using Phenix, followed by two rounds of model building in Coot (*Emsley et al., 2010*) and maximum-likelihood refinement in Phenix. The two EC4 domains were then manually placed in the electron density and subjected to rigid body refinement. Following a further two iterative rounds of model building and refinement the two EC1 domains were manually placed. Iterative model building and refinement continued yielding the final crystal structure whose statistics are given in *Figure 3—source data 1*.

## Structure analysis

Buried surface areas were calculated using 'Protein interfaces, surfaces and assemblies' service (PISA) at the European Bioinformatics Institute (http://www.ebi.ac.uk/pdbe/prot_int/pistart.html) (*Krissinel and Henrick, 2007*) and are given as the change in accessible surface area over both protomers. Root mean square deviations over aligned Cα atoms (RMSDs) between structures were calculated using Pymol (Schrödinger, LLC). Crystal structure figures were made using Pymol (Schrödinger, LLC).

## Sequence analysis

Multiple sequence alignments were generated using Clustal Omega (*Sievers et al., 2011*) and visualized using ESPript3.0 (*Robert and Gouet, 2014*). Sequence logos were generated from multiple sequence alignments using WebLogo3 (*Crooks et al., 2004*).

## Amino acid sequence alignment of cPcdhs γB7, γA4, and γC3 EC1–6 regions

CLUSTAL O(1.2.4) multiple sequence alignment

```
γB7    -QPVRYSIPEELDRGSVVGKLAKDLGLSVLEVSARKLRVS--AEKLHFSVDSESGDLLVK    57
γA4    -EQIRYSVPEELERGSVVGNLAADLGLEPGKLAERGVRIVSRGKTQLFALNPRSGSLVTA    59
γC3    STIIHYEILEERERGFPVGNVVTDLGLDLGSLSARRLRVVSGASRRFFEVNWETGEMFVN    60
       ::*.:  **  :**  **::.  ****.  .::  *  :*:    ..  *  ::  .:*.:..
γB7    DRIDREQICKGRRKCELQLEAVLENPLNIFHVVVEIEDVNDHAPQFPKDEINLEISESDS    117
γA4    GRVDREGLCDRSPKCTANLEILLEDKVRILAIEVEIIDVNDNAPSFGAQQREIKVAESEN    119
γC3    DRLDREELCGTLPSCTVTLELVVENPLELFSAEVVVQDINDNNPSFPTGEMKLEISEALA    120
       .*:*** :*    .* **  ::*: :.::  *  :  *:**:  *.*  :   : ::::  *:
γB7    PGARTILESAKDLDIGMNSLSKYQLSPNDYFLLLVKDNPDGSKYPELELQKMLDREAEST    177
γA4    PGTRFPLPEAFDLDIGVNALQGYQLSSNDHFSLDVQSGPDGIKYPELVLENALDREEEAV    179
γC3    PGTRFPLESAHDPDVGSNSLQTYELSHNEYFALRVQTREDGTKYAELVLERALDWEREPS    180
       **:*  *  .*  *  *:*  *:*.  *:**  *::*  *  *:  **  **  **  *:.  ** * *
γB7    HHLMLTAVDGGDPPRTGTTQLRIRVVDANDNRPVFSQDVYRVRLPEDLPPGTTVLRLKAM    237
γA4    HHLVLTAFDGGDPVRSGTATIQVTLVDTNDNAPVFTQPEYHISVKENLPVGTRLLTIKAT    239
γC3    VQLVLTALDGGTPARSATLPIRITVLDANDNAPAFNQSLYRARVREDAPPGTRVAQVLAT    240
       :*:***.*** *  *:.*  :::  ::*:***  *.*.*  *:  :  *:  * **  :  :  *
γB7    DQDEGINAEFTYSFLGV-ANK--AQFSLDPITGDIVTRQSLDFEEVEQYTIDVEAKDRGS    294
γA4    DPDEGVNGEVTYSFRNV-REKISQLFQLNSLTGDITVLGELDYEDSGFYDVDVEAHDGPG    298
γC3    DLDEGLNGEIVYSFGSHNRAGVRELFALDLVTGVLTIKGRLDFEDTKLHEIYIQAKDKGA    300
       *  ***:*.*.*.***.           *  *:  :**  :.    **:*:     :  :  ::*:*  .
γB7    --LSSQCKVIIEVLDENDNRPEIIITSLSDQISEDSPSGTVVALFKVRDRDSGENAEVMC    352
γA4    --LRARSKVLVTVLDVNDNAPEVTVTSLTSSIQEASSPGTVIALFNVHDSDSGENGLVTC    356
γC3    NPEGAHCKVLVEVVDVNDNAPEITVTSVYSPVPEDAPLGTVIALLSVTDLDAGENGLVTC    360
       ::.**:: *:* *** **: :**:  . : * : ***:**:.* * *:***. * *
γB7    SLSGNNPFKIHSSSNNYYKLVTDSILDREQTPGYNVTITATDRGKPPLSSSTTITLNVAD    412
γA4    SIPDNLPFRLEKTYGNYHRLLIHRTLDREEVSDYNITITATDQGTPPLSTETYISLQVVD    416
γC3    EVPPGLPFSLTSSLKNYFTLKTSAALDRETMPEYNLSITARDSGIPSLSALTTVKVQVSD    420
       .:  .  **  :  .:  **. *      ****    **::***  *  *  *  **:  *  :.::*  *
γB7    VNDNAPVFQQQAYLINVAENNQPGTSITQVKAWDPDVGSNGLVSYSIIASDLEPKALSSF    472
γA4    INDNPPTFTHASYSAYIPENNPRGASILSITAQDPDSGENAQVIYSLSEDTIQGAPMSSY    476
γC3    INDNPPQSSQSSYDVYVEENNLPGVPILNLSVWDPDAPPNARLSFFLLEPGAETGLVSRY    480
       :***  *  :  :*     :  ***  *.  *    .:...  ***  *.  :  :  :      :  :*  :
γB7    VSVNQDSGVVYAQRAFDHEQIRSFQLTLQARDQGSPALSANVSMRVLVDDRNDNAPRVLY    532
γA4    VSINSNTGVLYALRSFDYEQFQDLKLLVTARDSGTPPLSSNVSLSLSVLDQNDNTPEILY    536
γC3    FTINRDNGVLTTLVPLDYEDQREFQLTAHINDGGTPVLATNISVNVFVTDRNDNAPQVLY    540
       .::*  :.**:  :    :*:*:  :.::*     .*  *:*  *::*:*:  :  *  *:***:*.:**
γB7    PTLEPDGSALFDMVPRAAEPGYLVTKVVAVDADSGHNAWLSYHVLQASDPGLFSLGLRTG    592
γA4    PTIPTDGSTGVELTPRSADPGYLVTKVVAVDKDSGQNAWLSYRLLKASEPGLFSVGLHTG    596
γC3    PR---PGQSSVEMLPRGTAAGHVVSRVVGWDADAGHNAWLSYSLLGAPNQSLFAVGLHTG    597
```

```
*    *.: .:: **.: *::*::**. * *:*:****** :* * : .**::**:**
γB7    EVRTARALSDKDAARQRLLVAVRDGGQPPLSATATLLLVFADSLQE            638
γA4    EVRTARALLDRDALKQSLVVTVQDHGQPPLSATVTLTIAVSDNIPD            642
γC3    QISTARPIQDTDSPRQILTVLISDSGEPLLSTTATLTVSVTEESPE            643
:: *** : * *: :* * * : * *:* **:*.** : .::. :
```

## Structure-based sequence analysis of the γA4/γC3 interaction

Since both $\gamma A4_{3-6}$ and $\gamma C3_{3-6}$ are monomeric in solution but form a robust heterodimer when mixed (in SPR, AUC, and SEC-MALS) we hypothesized that these molecules might have opposing *cis* interaction side preferences. To facilitate hypothesis generation on the nature of their *cis* heterodimer interaction we modeled the two possible γA4/γC3 *cis* dimers: one with γA4 occupying the EC6-only position and γC3 the EC5–6 position; and the second with γC3 in the EC6-only position and γA4 in the EC5–6 position. To do this the monomeric $\gamma A4_{EC3-6}$ crystal structure (PDB: 5SZQ) was structurally superimposed over EC6 domains with the EC6-only protomer from the $\gamma B7_{EC3-6}$ *cis* dimer crystal structure (PDB: 5V5X; RMSD 0.7 Å over 91 aligned Cαs) or over EC5–6 domains with the EC5–6 protomer (RMSD 1.0 Å over 194 aligned Cαs). Since γA4 and γB7 are so structurally similar in their EC5–6 regions modeling γA4's *cis* interactions in this manner as a basis for hypothesis generation seemed reasonable. The only region of significant structural deviation within the EC5–6 regions between γA4 and γB7 is in the EC6 A–A' loop region which has a peripheral role in the EC6-only protomer interface. For modeling γC3 we used computational mutagenesis of the γB7 structure selecting the best-fit rotamer for each amino acid from the Dunbrack rotamer library (*Shapovalov and Dunbrack, 2011*), implemented in UCSF Chimera (*Pettersen et al., 2004*). No energy minimization was conducted and the models are intended only for use in hypothesis generation.

## *Cis* interface mutants

Our studies of Pcdh *cis* interactions we have found that mutagenesis of the *cis* interface commonly has a deleterious impact on protein expression levels in our system (*Goodman et al., 2017*). We assume this is because *cis* interaction is required for robust cell surface delivery/secretion (*Thu et al., 2014*), although this has not been specifically addressed in our HEK293 protein expression system.

To test our structure-guided hypotheses regarding γA4 and γC3s' *cis* interactions and side preferences as we tried to make a number of different *cis* interface mutants and were able to obtain four different mutants (see table below). Since protein yields were generally too low for AUC and SPR, MALS was used to study the impact of these mutants on γA4/γC3 *cis* dimer formation.

| Mutant protein (γB7 numbering given in parentheses) | *Cis* interface side targeted | Protein expression in 25 ml test |
|---|---|---|
| γC3 EC3–6 Y540G (Y532G equivalent) | EC6-only | No |
| γC3 EC3–6 V560D (L555D equivalent) | EC6-only | No |
| γC3 EC3–6 V565R (V560R equivalent) | EC6-only | Yes |
| γC3 EC3–6 A575R (A570R equivalent) | EC5–6 | No |
| γC3 EC3–6 R563K (K558R equivalent) | Both | Yes |
| γA4 EC3–6 Y536G (Y532G equivalent) | EC6-only | No |
| γA4 EC3–6 L559D (L555D equivalent) | EC6-only | No |
| γA4 EC3–6 V564R (V560R equivalent) | EC6-only | Yes |
| γA4 EC3–6 A574R (A570R equivalent) | EC5–6 | No |
| γA4 EC3–6 K562R (K558R equivalent) | EC6-only | Yes |
| β1 EC3–6 V563R (V560R equivalent) | EC6-only | No |
| β1 EC3–6 S573R (A570R equivalent) | EC6-only | No |
| β1 EC3–6 K561R (K558R equivalent) | EC5–6 | No |

*Continued on next page*

*Continued*

| Mutant protein (γB7 numbering given in parentheses) | *Cis* interface side targeted | Protein expression in 25 ml test |
|---|---|---|
| β9 EC3–6 V563R (V560R equivalent) | EC6-only | No |
| β9 EC3–6 A573R (A570R equivalent) | EC6-only | No |
| β9 EC3–6 K561R (K558R equivalent) | EC5–6 | No |

## Acknowledgements

We thank Surajit Banerjee for help with synchrotron data collection at the APS NE-CAT 24-ID-C/E beam-lines, supported by NIH P41GM103403. This work was supported by the NIH (grants R01MH114817 to LS and R01DK106548 to RS), the National Science Foundation (grant MCB-1914542 to BH), the Israel Science Foundation (grant 1463/19 to RR), and the Israel Cancer Research Fund (grant ICRF 19-203-RCDA to RR).

## Additional information

### Funding

| Funder | Grant reference number | Author |
|---|---|---|
| National Institutes of Health | R01MH114817 | Lawrence Shapiro |
| National Institutes of Health | R01DK106548 | Rosemary V Sampogna |
| National Science Foundation | MCB-1914542 | Barry Honig |
| Israel Science Foundation | 1463/19 | Rotem Rubinstein |
| Israel Cancer Research Fund | ICRF19-203-RCDA | Rotem Rubinstein |

The funders had no role in study design, data collection, and interpretation, or the decision to submit the work for publication.

### Author contributions

Kerry Marie Goodman, Conceptualization, Data curation, Formal analysis, Writing – original draft, Writing – review and editing, Cloned, expressed, purified and crystallized proteins; Phinikoula S Katsamba, Data curation, Formal analysis, Methodology, Writing – review and editing, Writing – original draft, Performed and analyzed SPR experiments; Rotem Rubinstein, Writing – review and editing; Göran Ahlsén, Data curation, Formal analysis, Performed and analyzed the analytical ultracentrifugation and multi-angle light scattering experiments; Fabiana Bahna, Data curation, Formal analysis, Cloned, expressed, purified and crystallized the proteins; Seetha Mannepalli, Data curation, Formal analysis, Cloned, expressed, purified and crystallized the proteins; Hanbin Dan, Data curation, Formal analysis, Performed and analyzed the cell aggregation experiments; Rosemary V Sampogna, Data curation, Formal analysis, Funding acquisition, Performed and analyzed the cell aggregation experiments; Lawrence Shapiro, Conceptualization, Funding acquisition, Writing – original draft, Writing – review and editing, Formal analysis, Designed experiments, analyzed data, drafted and edited the manuscript; Barry Honig, Conceptualization, Writing – original draft, Writing – review and editing, Formal analysis, Funding acquisition, Designed experiments, analyzed data, drafted and edited the manuscript

### Author ORCIDs

Kerry Marie Goodman (iD) http://orcid.org/0000-0003-2063-5823
Phinikoula S Katsamba (iD) http://orcid.org/0000-0003-3981-1604

Rotem Rubinstein http://orcid.org/0000-0003-3657-5984
Rosemary V Sampogna http://orcid.org/0000-0002-1279-4552
Lawrence Shapiro http://orcid.org/0000-0001-9943-8819
Barry Honig http://orcid.org/0000-0002-2480-6696

**Decision letter and Author response**
Decision letter https://doi.org/10.7554/eLife.72416.sa1
Author response https://doi.org/10.7554/eLife.72416.sa2

## Additional files

### Supplementary files
• Transparent reporting form

### Data availability
Atomic coordinates and structure factors have been deposited in the PDB under the accession codes 7JGZ and 7RGF.

The following datasets were generated:

| Author(s) | Year | Dataset title | Dataset URL | Database and Identifier |
|---|---|---|---|---|
| Goodman KM | 2021 | Protocadherin gammaC4 EC1-4 crystal structure | https://www.rcsb.org/structure/7JGZ | RCSB Protein Data Bank, 7JGZ |
| Goodman KM | 2021 | Protocadherin gammaC4 EC1-4 crystal structure | https://www.rcsb.org/structure/7RGF | RCSB Protein Data Bank, 7RGF |

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

## Appendix 1

### Appendix 1—key resources table

| Reagent type (species) or resource | Designation | Source or reference | Identifiers | Additional information |
|---|---|---|---|---|
| Strain, strain background (*E. coli*) | One shot Top10 Competent Cells | Invitrogen | C4040-06 | Plasmid production |
| Cell line (*Homo sapiens*) | FreeStyle 293 F cells | Thermo Fisher Scientific | R79007 | Cell line for protein expression |
| Cell line (*Homo sapiens*) | K-562 bone marrow chronic myelogenous leukemia (CML) cells | ATCC | ATCC CCL-243 | Cell line for cell-aggregation assays |
| Transfected construct (*M. musculus*) | $\alpha4_{1-5}$ | This paper | | Pcdhα4 EC1–5, Honig/Shapiro labs |
| Transfected construct (*M. musculus*) | $\alpha7_{1-5}$ | *Rubinstein et al., 2015* | | |
| Transfected construct (*M. musculus*) | $\alpha12_{1-5}$ | This paper | | Pcdhα12 EC1–5, Honig/Shapiro labs |
| Transfected construct (*M. musculus*) | $\beta6_{1-4}$ | *Goodman et al., 2016c* | | |
| Transfected construct (*M. musculus*) | $\beta8_{1-4}$ | *Goodman et al., 2016c* | | |
| Transfected construct (*M. musculus*) | $\gamma A1_{1-4}$ | *Goodman et al., 2016a* | | |
| Transfected construct (*M. musculus*) | $\gamma A4_{1-4}$ | *Goodman et al., 2016a* | | |
| Transfected construct (*M. musculus*) | $\gamma A8_{1-4}$ | *Rubinstein et al., 2015* | | |
| Transfected construct (*M. musculus*) | $\gamma A9_{1-5}$ | *Goodman et al., 2016a* | | |
| Transfected construct (*M. musculus*) | $\gamma B2_{1-5}$ | *Goodman et al., 2016a* | | |
| Transfected construct (*M. musculus*) | $\gamma B4_{1-5}$ | This paper | | PcdhγB4 EC1–5, Honig/Shapiro labs |
| Transfected construct (*M. musculus*) | $\gamma B5_{1-4}$ | *Goodman et al., 2016a* | | |
| Transfected construct (*M. musculus*) | $\alpha C2_{1-4}$ | *Rubinstein et al., 2015* | | |
| Transfected construct (*M. musculus*) | $\gamma C3_{1-4}$ | *Goodman et al., 2016a* | | |
| Transfected construct (*M. musculus*) | $\gamma C4_{1-4}$ | This paper | | PcdhγC4 EC1–4, Honig/Shapiro labs |

*Appendix 1 Continued on next page*

Appendix 1 Continued

| Reagent type (species) or resource | Designation | Source or reference | Identifiers | Additional information |
|---|---|---|---|---|
| TRansfected construct (*M. musculus*) | $\gamma C5_{1-5}$ | **Rubinstein et al., 2015** | | |
| Transfected construct (*M. musculus*) | $\alpha 7_{1-5}$-AVI | This paper | | Biotinylated Pcdhα7 EC1–5, Honig/Shapiro labs |
| Transfected construct (*M. musculus*) | $\beta 6_{1-4}$-AVI | This paper | | Biotinylated Pcdhβ6 EC1–4, Honig/Shapiro labs |
| Transfected construct (*M. musculus*) | $\beta 8_{1-4}$-AVI | This paper | | Biotinylated Pcdhβ8 EC1–4, Honig/Shapiro labs |
| Transfected construct (*M. musculus*) | $\gamma A8_{1-4}$-AVI | This paper | | Biotinylated PcdhγA8 EC1–4, Honig/Shapiro labs |
| Transfected construct (*M. musculus*) | $\gamma A9_{1-5}$-AVI | This paper | | Biotinylated PcdhγA9 EC1–5, Honig/Shapiro labs |
| Transfected construct (*M. musculus*) | $\gamma B2_{1-5}$-AVI | This paper | | Biotinylated PcdhγB2 EC1–5, Honig/Shapiro labs |
| Transfected construct (*M. musculus*) | $\alpha C2_{1-4}$-AVI | This paper | | Biotinylated PcdhαC2 EC1–4, Honig/Shapiro labs |
| Transfected construct (*M. musculus*) | $\gamma C3_{1-4}$-AVI | This paper | | Biotinylated PcdhγC3 EC1–4, Honig/Shapiro labs |
| Transfected construct (*M. musculus*) | $\gamma C4_{1-4}$-AVI | This paper | | Biotinylated PcdhγC4 EC1–4, Honig/Shapiro labs |
| Transfected construct (*M. musculus*) | $\gamma C5_{1-5}$-AVI | This paper | | Biotinylated PcdhγC5 EC1–5, Honig/Shapiro labs |
| Transfected construct (*M. musculus*) | $\alpha 4_{1-4}$-AVI | This paper | | Biotinylated Pcdhα4 EC1–4, Honig/Shapiro labs |
| Transfected construct (*M. musculus*) | $\alpha 7_{1-5}$ L301R | This paper | | Pcdhα7 EC1–5 mutant, Honig/Shapiro labs |
| Transfected construct (*M. musculus*) | $\gamma A8_{1-4}$ I116R | **Rubinstein et al., 2015** | | PcdhγA8 EC1–4 mutant, Honig/Shapiro labs |
| Transfected construct (*M. musculus*) | $\beta 6_{1-4}$ R41N | This paper | | Pcdhβ6 EC1–4 mutant, Honig/Shapiro labs |
| Transfected construct (*M. musculus*) | $\gamma C5_{1-5}$ S116R | This paper | | PcdhγC5 EC1–5 mutant, Honig/Shapiro labs |
| Transfected construct (*M. musculus*) | $\beta 6_{1-4}$ S117I | This paper | | Pcdhβ6 EC1–4 mutant, Honig/Shapiro labs |
| Transfected construct (*M. musculus*) | $\beta 6_{1-4}$ L125P | This paper | | Pcdhβ6 EC1–4 mutant, Honig/Shapiro labs |
| Transfected construct (*M. musculus*) | $\beta 6_{1-4}$ E369K | This paper | | Pcdhβ6 EC1–4 mutant, Honig/Shapiro labs |

*Appendix 1 Continued on next page*

*Appendix 1 Continued*

| Reagent type (species) or resource | Designation | Source or reference | Identifiers | Additional information |
|---|---|---|---|---|
| Transfected construct (*M. musculus*) | β6$_{1-4}$ Y371F | This paper | | Pcdhβ6 EC1–4 mutant, Honig/Shapiro labs |
| Transfected construct (*M. musculus*) | β6$_{1-4}$ R41N/S117I | This paper | | Pcdhβ6 EC1–4 mutant, Honig/Shapiro labs |
| Transfected construct (*M. musculus*) | β6$_{1-4}$ R41N/E369K | This paper | | Pcdhβ6 EC1–4 mutant, Honig/Shapiro labs |
| Transfected construct (*M. musculus*) | β6$_{1-4}$ S117I/L125P | This paper | | Pcdhβ6 EC1–4 mutant, Honig/Shapiro labs |
| Transfected construct (*M. musculus*) | β6$_{1-4}$ R41N/S117I/L125P | This paper | | Pcdhβ6 EC1–4 mutant, Honig/Shapiro labs |
| Transfected construct (*M. musculus*) | β6$_{1-4}$ R41N/S117I/E369K | This paper | | Pcdhβ6 EC1–4 mutant, Honig/Shapiro labs |
| Transfected construct (*M. musculus*) | β6$_{1-4}$ R41N/S117I/Y371F | This paper | | Pcdhβ6 EC1–4 mutant, Honig/Shapiro labs |
| Transfected construct (*M. musculus*) | β6$_{1-4}$ R41N/S117I/L125P/E369K/Y371F | This paper | | Pcdhβ6 EC1–4 mutant, Honig/Shapiro labs |
| Transfected construct (*M. musculus*) | γC4$_{1-4}$ E78A | This paper | | PcdhγC4 EC1–4 mutant, Honig/Shapiro labs |
| Transfected construct (*M. musculus*) | γC4$_{1-4}$ E78Q | This paper | | PcdhγC4 EC1–4 mutant, Honig/Shapiro labs |
| Transfected construct (*M. musculus*) | γC4$_{1-4}$ S344R | This paper | | PcdhγC4 EC1–4 mutant, Honig/Shapiro labs |
| Transfected construct (*M. musculus*) | γC4$_{1-4}$ D290A | This paper | | PcdhγC4 EC1–4 mutant, Honig/Shapiro labs |
| Transfected construct (*M. musculus*) | γC4$_{1-4}$ D290N | This paper | | PcdhγC4 EC1–4 mutant, Honig/Shapiro labs |
| Transfected construct (*M. musculus*) | β1$_{3-6}$ | This paper | | Pcdhβ1 EC3–6, Honig/Shapiro labs |
| Transfected construct (*M. musculus*) | β6$_{1-6}$ | This paper | | Pcdhβ6 EC1–6, Honig/Shapiro labs |
| Transfected construct (*M. musculus*) | β9$_{3-6}$ | This paper | | Pcdhβ9 EC3–6, Honig/Shapiro labs |
| Transfected construct (*M. musculus*) | γA3$_{3-6}$ | This paper | | PcdhγA3 EC3–6, Honig/Shapiro labs |
| Transfected construct (*M. musculus*) | γA4$_{3-6}$ | *Goodman et al., 2016a* | | |
| Transfected construct (*M. musculus*) | γA9$_{3-6}$ | This paper | | PcdhγA9 EC3–6, Honig/Shapiro labs |

*Appendix 1 Continued on next page*

*Appendix 1 Continued*

| Reagent type (species) or resource | Designation | Source or reference | Identifiers | Additional information |
|---|---|---|---|---|
| Transfected construct (*M. musculus*) | γB2$_{3-6}$ | **Goodman et al., 2016a** | | |
| Transfected construct (*M. musculus*) | γB5$_{3-6}$ | **Goodman et al., 2016a** | | |
| Transfected construct (*M. musculus*) | γB7$_{3-6}$ | **Goodman et al., 2016a** | | |
| Transfected construct (*M. musculus*) | αC2$_{2-6}$ | **Goodman et al., 2016a** | | |
| Transfected construct (*M. musculus*) | α7$_{1-5}$/γC3$_6$ chimera | **Goodman et al., 2016a** | | |
| Transfected construct (*M. musculus*) | γC3$_{3-6}$ | **Goodman et al., 2016a** | | |
| Transfected construct (*M. musculus*) | γC5$_{2-6}$ | This paper | | PcdhγC5 EC2–6, Honig/Shapiro labs |
| Transfected construct (*M. musculus*) | β9$_{3-6}$-AVI | This paper | | Biotinylated Pcdh β9 EC3–6, Honig/Shapiro labs |
| Transfected construct (*M. musculus*) | γA4$_{3-6}$-AVI | This paper | | Biotinylated PcdhγA4 EC3–6, Honig/Shapiro labs |
| Transfected construct (*M. musculus*) | γA9$_{3-6}$-AVI | This paper | | Biotinylated PcdhγA9 EC3–6, Honig/Shapiro labs |
| Transfected construct (*M. musculus*) | γB2$_{3-6}$-AVI | This paper | | Biotinylated PcdhγB2 EC3–6, Honig/Shapiro labs |
| Transfected construct (*M. musculus*) | αC2$_{3-6}$-AVI | This paper | | Biotinylated Pcdh αC2 EC3–6, Honig/Shapiro labs |
| Transfected construct (*M. musculus*) | γC3$_{3-6}$-AVI | This paper | | Biotinylated PcdhγC3 EC3–6, Honig/Shapiro labs |
| Transfected construct (*M. musculus*) | γC5$_{2-6}$-AVI | This paper | | Biotinylated PcdhγC5 EC2–6, Honig/Shapiro labs |
| Transfected construct (*M. musculus*) | γA4$_{3-6}$ V560R | This paper | | PcdhγA4 EC3–6 mutant, Honig/Shapiro labs |
| Transfected construct (*M. musculus*) | γC3$_{3-6}$ V560R | This paper | | PcdhγC3 EC3–6 mutant, Honig/Shapiro labs |
| Transfected construct (*M. musculus*) | γA4$_{3-6}$ K558R | This paper | | PcdhγA4 EC3–6 mutant, Honig/Shapiro labs |
| Transfected construct (*M. musculus*) | γC3$_{3-6}$ R558K | This paper | | PcdhγC3 EC3-6 mutant, Honig/Shapiro labs |
| Transfected construct (*M. musculus*) | γB7$_{3-6}$ Y532G | **Goodman et al., 2017** | | |

*Appendix 1 Continued on next page*

*Appendix 1 Continued*

| Reagent type (species) or resource | Designation | Source or reference | Identifiers | Additional information |
|---|---|---|---|---|
| Transfected construct (*M. musculus*) | γB7$_{3–6}$ A570R | This paper | | PcdhγB7 EC3–6 mutant, Honig/Shapiro labs |
| Peptide, recombinant protein | NeutrAvidin-HRP | Thermo Fisher Scientific | 31,030 | Biotinylated protein western bot |
| Peptide, recombinant protein | NeutrAvidin protein | Thermo Fisher Scientific | 31,000 | SPR assays |
| Peptide, recombinant protein | BSA | Sigma-Aldrich | A7906 | SPR assays |
| Commercial assay or kit | Spin Miniprep Kit | Qiagen | 27,106 | |
| Commercial assay or kit | Hi-speed Plasmid Maxi Kit | Qiagen | 12,663 | |
| Commercial assay or kit | SF Cell Line 4D-Nucleofector X Kit S | Lonza | V4XC-2032 | |
| Commercial assay or kit | Amine-coupling kit | Cytiva | BR100050 | SPR experiments |
| Commercial assay or kit | Morpheus Amino Acids | Molecular Dimensions | MD2-100-77 | Crystallography |
| Commercial assay or kit | Morpheus Buffer System II | Molecular Dimensions | MD2-100-101 | Crystallography |
| Chemical compound | Polyethylenimine | Polysciences | 24765-2 | Transfection |
| Chemical compound | Biotin | Sigma-Aldrich | B4501 | Protein biotinylation |
| Chemical compound | Tris Base | Fisher Scientific | BP152-5 | |
| Chemical compound | Sodium Chloride | Fisher Scientific | S271-10 | |
| Chemical compound | Calcium Chloride Dihydrate | JT Baker | 1336-01 | |
| Chemical compound | Imidazole | ACROS | 301870025 | |
| Chemical compound | HEPES | Sigma-Aldrich | H3375 | |
| Chemical compound | Tween-20 | Sigma-Aldrich | P7949 | |
| Chemical compound | Sodium Acetate | Sigma-Aldrich | S7545 | |
| Chemical compound | IMAC Sepharose 6 Fast Flow | Cytiva | 17092109 | |
| Chemical compound | Penicillin Streptomycin | Thermo Fisher Scientific | 15070063 | |
| Chemical compound | PEG 6000 | Sigma-Aldrich | 81,260 | |
| Chemical compound | PEG 8000 | Sigma-Aldrich | 89,510 | |
| Chemical compound | Ethylene Glycol | Fluka | 03760 | |
| Chemical compound | Lithium Chloride | Sigma-Aldrich | L8895 | |
| Chemical compound | MES | Sigma-Aldrich | M3671 | |
| Chemical compound | Glycerol | ACROS | 332031000 | |
| Software, algorithm | UCLA Diffraction Anisotropy Server | **Strong et al., 2006** | | https://srv.mbi.ucla.edu/Anisoscal/ |
| Software, algorithm | SednTerp | Thomas Laue | | http://bitcwiki.sr.unh.edu/index.php/Main_Page |
| Software, algorithm | HeteroAnalysis | | | https://core.uconn.edu/auf |
| Software, algorithm | Scrubber 2.0 | BioLogic Software | | http://www.biologic.com.au |

*Appendix 1 Continued on next page*

*Appendix 1 Continued*

| Reagent type (species) or resource | Designation | Source or reference | Identifiers | Additional information |
|---|---|---|---|---|
| Software, algorithm | Phaser | *McCoy et al., 2007* | | Implemented in CCP4 or Phenix (see below) |
| Software, algorithm | CCP4 | *Winn et al., 2011* | | https://www.ccp4.ac.uk/ |
| Software, algorithm | Phenix | *Liebschner et al., 2019* | | http://www.hkl-xray.com/ |
| Software, algorithm | XDS | *Kabsch, 2010* | | http://xds.mpimf-heidelberg.mpg.de |
| Software, algorithm | AIMLESS | *Evans and Murshudov, 2013* | | http://www.ccp4.ac.uk |
| Software, algorithm | Coot | *Emsley et al., 2010* | | https://www2.mrc-lmb.cam.ac.uk/personal/pemsley/coot/ |
| Software, algorithm | PISA | *Krissinel and Henrick, 2007* | | http://www.ebi.ac.uk/pdbe/protint/pistart.html |
| Software, algorithm | Pymol | Schrödinger | | https://pymol.org |
| Software, algorithm | UCSF Chimera | *Pettersen et al., 2004* | | https://www.cgl.ucsf.edu/chimera/ |
| Software, algorithm | Clustal Omega | *Sievers et al., 2011* | | https://www.ebi.ac.uk/Tools/msa/clustalo/ |
| Software, algorithm | WebLogo 3.0 | *Crooks et al., 2004* | | http://weblogo.threeplusone.com/ |
| Software, algorithm | SignalP 4.0 | *Petersen et al., 2011* | | https://services.healthtech.dtu.dk/service.php?SignalP-5.0 |
| Software, algorithm | ASTRA | Wyatt | | https://www.wyatt.com/products/software/astra.html |
| Other | Freestyle 23 Expression Media | Thermo Fisher Scientific | 12338-018 | Protein expression media |
| Other | Opti-MEM Reduced Serum Media | Thermo Fisher Scientific | 31985-070 | Protein expression media |
| Other | Series S CM4 chip | Cytiva | BR100539 | SPR assays |
| Other | Fetal Bovine Serum | Thermo Fisher Scientific | 16141079 | Cell-aggregation assays media |
| Other | DMEM with GlutaMAX | Thermo Fisher Scientific | 10569010 | Cell-aggregation assays media |

