## [Editor Report]

The direct investigation of homotypic and heterotypical preference between cis and trans interactions among the protocadherin isoforms is an important step to understand the mechanisms of self avoidance. We are particularly excited about the discovery that the discovery that showed cis interactions are promiscuous, but with preferences favoring formation of heterologous cis dimers. Trans-homophilic interactions are remarkably precise, with no evidence for heterophilic interactions between different isoforms.

---

## [Decision Letter]

**Decision letter after peer review:**

Thank you for submitting your article "How clustered protocadherin binding specificity is tuned for neuronal self/non-self-recognition" for consideration by *eLife*. Your article has been reviewed by 3 peer reviewers, and the evaluation has been overseen by a Reviewing Editor and Olga Boudker as the Senior Editor. The following individual involved in review of your submission has agreed to reveal their identity: Rachelle Gaudet (Reviewer #2).

Essential revisions:

1) The reviewers all agreed that additional mutagenesis on E78 and D290 followed by SRM would further.

2) Address other points raised by the three reviewers with discussion and clarifications.

*Reviewer #1 (Recommendations for the authors):*

Overall, the manuscript sets out to answer a specific, limited set of questions, and does that very well. The limited nature of the questions asked may be an issue for *eLife*, but that's not a question for me to respond to, especially when no simple culture system to test these ideas is readily available (cell aggregation assays, while immensely useful, may not be a complete proxy for self-recognition and synapse avoidance).

1. The MALS results, important for conclusions on which side of the cis dimer cPcdhs prefer, is a bit confusing. In Figure 5B, dimers are running at at intermediate molar mass. Can the authors explain why? Does this mean the peaks labeled dimer are a mixture with roughly 50% monomer? Since the third panel shows good separation between monomer and dimer, I am not convinced that is the explanation.

2. The extensive use of the word "alternate", while appropriate here and common in cPcdh literature, may be effecting readability. For example, see "the alternate and other C-type cPcdhs". Does alternate include only the same class or all cPcdhs? Please review and ignore if no issues are found.

3. Please better explain the statement "We estimate that, even for these pairs, the lower limit for KD would be ~200 μM." Which pairs?

4. Similarly, "Strict homophilic specificity is therefore not contingent on the strength of the homophilic interaction" is a confusing statement. I believe the authors simply want to state that trans homodimers come with KDs with a >200-fold range, while still working to effectively form trans dimers between cells in aggregation assays. This is more of a statement about affinity, rather than specificity.

5. The small differences between the two structures of the gammaC4 dimer is interesting. I do not think that the mentioned pH/histidine protonation is a convincing explanation, especially given the atomic details of those sites (as depicted in Fig. 3 suppl 1 B-ii). Wouldn't a dynamic, low affinity interaction be a more plausible explanation, as seen by MD before in Nicoludis et al. 2019 and by crystallography in Nicoludis et al. 2016? Crystal lattice forces can overcome such transitionary, weak interactions. The high-resolution structure reported here may support, and be supported by, those studies.

6. The models in Figure 5 suppl 1: Could you expand a bit more in the Methods on model generation? Simple mutagenesis of side chain with the most likely/least clashing rotamers picked? Was any energy minimization employed? (The disclaimer in the legend for non-structural biologists/biochemists, and including a methods section for hypothesis generation is very much appreciated.)

7. The authors state that "zippers consisting of homodimers are easier to form in a kinetic sense than heterodimeric zippers ..." They need to better explain this statement for the clarity of their argument.

*Reviewer #2 (Recommendations for the authors):*In two instances in the first page of the introduction the authors erroneously generalize to "vertebrates" the fact that cPcdhs are in three gene clusters. Fish do not have β clusters, but rather have 2 each of the α and γ clusters, and *Xenopus* tropicalis also lacks a β cluster (https://pubmed.ncbi.nlm.nih.gov/27261006/). "Mammals" would be an appropriate substitution.

Regarding the presentation of the data in Figure 2, several of the isoforms show now interactions at all (α121-5, γA41-4, γB51-4). Later in the manuscript, AUC data are introduced that demonstrate that all three of these proteins do form homodimers. This would be worth mentioning on p. 8, as it alleviates potential worries about protein quality.

The result section on the γC41-4 structures is underwhelming. First, the RMSD comparisons and correlations to sequence similarity provide little additional useful information. Second, the attempt at an explanation of the low homodimerization affinity of γC41-4 does not seem to hold up upon inspection of the structures or Figure 3—figure supplement 2. The authors try to argue that the low affinity may be due to "two buried charges in the interface, E78 and D290." E78 is in proximity to R89 and in some of the three structural instances forms a (longish) hydrogen bond to S344 across the interface, and other instances it is not quite buried. Similarly, D290 is hardly buried, and in proximity to K127 and R156 across the interface and H255 and H304 on the same subunit. Looking at the electrostatic surfaces of the interface similarly does not suggest a strong, if any, electrostatic mismatch.

The experiments, using mutagenesis, to investigate the asymmetry of cis dimerization are very interesting. It is a bit disappointing that the authors do not present a gain-of-function experiments – all the mutations are loss-of-function. Although the fact that the mutations lead to selective loss-of-function of specific interfaces do provide quite strong support for the inferences the authors make. One exception is this statement: "SPR data for the γB7 mutants over the αC2 surface suggests αC2 preferentially occupies the EC6-only side in αC2/γB7 dimers". The experiment is a negative result, showing, using a Y-to-G mutation, that Y532 is not essential to the potential EC5-6 interface. So while the authors do wisely use the word "suggests", the above caveat should be presented as well.

Can the authors speculate as to why the trans interactions have evolved to be so specific?

On p. 16, lines 23-26, the sentence is confusing and should be rephrased. Are the authors suggesting that all cPcdhs will form dimers when constrained to a 2D membrane surface? If so, the use of the expression "monomeric cPcdhs" is confusing (i.e. do the authors mean cPcdhs that fail to dimerize in solution, or truly "monomeric" cPcdhs?).

*Reviewer #3 (Recommendations for the authors):*

The authors may consider adding experiments that would extend the new conclusions drawn. One of the most interesting analyses is presented in Figure 3—figure supplement 2, where the structures of the C4 and B2 trans dimer interfaces are compared. It is suggested that C4 residue E78 may be destabilizing; could the authors continue their fruitful mutational approach here and determine if this is the case? The same could be done for D290 in C4. Mutating these residues or any adjacent ones so that they match those of gB2 or a delta2 Pcdh could be informative.

Similarly, given what the authors have admirably shown regarding the mechanism of cis interaction and some of the residues that are crucial for the EC5/6 vs EC6 side of the dimer, can they utilize the data in Figure 4—figure supplement 3 to make predictions about EC5/6 vs EC6 "handedness" preference and test a few further examples to determine if this is likely to be generally predictable from sequence alone? If so then an interaction matrix could be determined that would help drive predictions in functional studies.

The Discussion section could be improved by more clearly detailing how the new results here may explain, or are either consistent or inconsistent with, prior functional studies. For example, C3 does not homodimerize in cis (or at least does not do so preferentially). And yet in Lefebvre et al., 2012, re-introducing only C3 in a γ cluster knock out rescued self-avoidance in starburst amacrine cells while in Molumby et al., 2016 expression of C3 in cortical neurons on a knockout background rescued and actually enhanced dendrite arbor complexity. Similarly, alphaC2 is able to solely effect axonal tiling in some neurons (Chen et al., 2017). How do the authors envision the preferences for interfamily cis dimerization playing out functionally then? It is stated on the third page of the discussion (there are no page numbers), lines 6-10, that carrier isoforms may be able to fill both the EC6 and EC5/6 roles. If this is so then what would the importance of the interaction preferences so nicely shown here be? Some more exploration of this, perhaps with some predictions to test in future functional studies, would be most helpful. On that same page, final paragraph, it is noted that weak homophilic cis interactions may indicate that once in the membrane, isoforms re-sort to allow for heterodimeric zippers that are a prerequisite for the chain termination model. But again, if in several cases expression of a single isoform can rescue KO phenotypes, perhaps the model needs to be amended? It is worth exploring that more deeply here.

Similarly, C4 does appear to interact in trans similarly to other cPcdhs, so what might explain its special role in maintaining viability and neuronal survival? Do the authors see this as likely mediated by the unique cytoplasmic region, or does the weak trans interaction suggest possible cis interactions with non Pcdh molecules such as Ret or ROR or neuroligins, etc?

Some suggestions for figures:

The main figure 3 is very slight, with only panel A anything new (panel B is unnecessary). Data from Figure 3—figure supplement 1B and 2B should be moved into the main figure.

Figure 2—figure supplement 1: in panel B, the experiment with alphaC2 shown on top is not mentioned anywhere in the paper, nor is it explained in the legend. What is being tested in that experiment and what is the conclusion?

Figure 4B: is ND ‘not determined’ or ‘not detected’? (if the latter, how does that differ from “>50”?). The legends in general are sparse and could be rewritten to explain the figures more.

A few writing suggestions:

The Introduction section could be better organized. There is repetition of information between the first and third paragraphs. There are also quite a few sentences that have extra or missing words or are otherwise ungrammatical. For example, on page 2 of the introduction, lines 29-30; this sentence should be rewritten. Line 30 of the section titled “Clustered protocadherin cis interactions are promiscuous with a range of interaction strengths” has an extra word “which”. There are other instances, please proofread for such grammatical errors/typos.

SPR and AUC should be defined at first usage.

The language used when explaining the preferential cis dimerization affinities for various cPcdhs could be more consistent or at least explained in more detail. While α Pcdhs are assumed to “only” be able to be an EC 5-6 donor, C3 is thought to “preferentially” be an EC5-6 donor. Since C3 does not dimerize in solution, wouldn’t this indicate that C3 can “only” be a 5-6 EC donor? Is it known which other EC3-6 constructs can self-dimerize, and if so wouldn’t that indicate which molecules can be considered “preferential” donors, while molecules which don’t self-dimerize at all are “only” one kind of donor? Perhaps the authors could discuss.

---

## [Author Response]

Essential revisions:1) The reviewers all agreed that additional mutagenesis on E78 and D290 followed by SRM would further.

We have carried out the requested experiments as described in the response to Reviewer 2. The experiments confirmed the role of E78 in reducing the binding affinity of cPcdh gC4 but not of D290. Relevant text has been added.

Reviewer #1 (Recommendations for the authors):Overall, the manuscript sets out to answer a specific, limited set of questions, and does that very well. The limited nature of the questions asked may be an issue for eLife, but that's not a question for me to respond to, especially when no simple culture system to test these ideas is readily available (cell aggregation assays, while immensely useful, may not be a complete proxy for self-recognition and synapse avoidance).1. The MALS results, important for conclusions on which side of the cis dimer cPcdhs prefer, is a bit confusing. In Figure 5B, dimers are running at at intermediate molar mass. Can the authors explain why? Does this mean the peaks labeled dimer are a mixture with roughly 50% monomer? Since the third panel shows good separation between monomer and dimer, I am not convinced that is the explanation.

During SEC-MALS experiments, a dimer/monomer equilibrium is established, as mixtures of γC3 and γA4 *cis* fragments (or their mutants) move through the size-exclusion chromatography column, which is dependent on the K_D_ of the interaction. For the wt γC3/γA4 interaction the peak labeled as dimer is a mixture of heterodimer and monomer populations in equilibrium. Although the concentrations of 18 μM for each γC3 and γA4 proteins are above the K_D_ of ~3 μM for the γC3/γA4 *cis* interaction, they are not sufficiently high for all the *cis* fragments to be bound into heterodimers, leaving a significant population of molecules as monomers, resulting in mws of ~76kDa for the dimeric peak compared to the predicted molecular weight for a dimer of ~108kDa. In the third panel (γA4+ γC3 V560R), the dimer and monomer peaks resolve completely, probably due to a lower K_D_, so at 18 μM of each fragment, only a small proportion of molecules can assemble into dimers, allowing the dimer and monomer peaks to elute at distinctly different volumes and therefore resolve. We have now included an explanation in the methods section to help the reader better understand the results of these experiments.

2. The extensive use of the word "alternate", while appropriate here and common in cPcdh literature, may be effecting readability. For example, see "the alternate and other C-type cPcdhs". Does alternate include only the same class or all cPcdhs? Please review and ignore if no issues are found.

Alternate cPcdhs includes all α, β, and γ cPcdhs except the C-types. We have added an additional sentence to the introduction to clarify the definition.

3. Please better explain the statement "We estimate that, even for these pairs, the lower limit for KD would be ~200 μM." Which pairs?

The sentence has now been reworded to read “We estimate that, for heterophilic trans-dimers, the lower limit for the dissociation constant (K_D_) would be ~200 μM”.

4. Similarly, "Strict homophilic specificity is therefore not contingent on the strength of the homophilic interaction" is a confusing statement. I believe the authors simply want to state that trans homodimers come with KDs with a >200-fold range, while still working to effectively form trans dimers between cells in aggregation assays. This is more of a statement about affinity, rather than specificity.

Thanks! We have clarified the relevant statement in the manuscript.

5. The small differences between the two structures of the gammaC4 dimer is interesting. I do not think that the mentioned pH/histidine protonation is a convincing explanation, especially given the atomic details of those sites (as depicted in Fig. 3 suppl 1 B-ii). Wouldn't a dynamic, low affinity interaction be a more plausible explanation, as seen by MD before in Nicoludis et al. 2019 and by crystallography in Nicoludis et al. 2016? Crystal lattice forces can overcome such transitionary, weak interactions. The high-resolution structure reported here may support, and be supported by, those studies.

We have amended the section to reflect the possibility that the two structures represent different conformational states of the dimer and referenced the Nicoludis et al. papers.

6. The models in Figure 5 suppl 1: Could you expand a bit more in the Methods on model generation? Simple mutagenesis of side chain with the most likely/least clashing rotamers picked? Was any energy minimization employed? (The disclaimer in the legend for non-structural biologists/biochemists, and including a methods section for hypothesis generation is very much appreciated.)

We have added further details and disclaimers to the legend and methods section. The models of gC3 were generated using simple mutagenesis of the gB7 cis dimer structure with most likely/best fit rotamer selection from the Dunbrack library. The monomeric structure of gA4 EC3–6 was used to model gA4 No energy minimization was employed. These models were solely intended for hypothesis generation purposes as it now states in the methods/figure legend.

7. The authors state that "zippers consisting of homodimers are easier to form in a kinetic sense than heterodimeric zippers ..." They need to better explain this statement for the clarity of their argument.

The argument as presented was indeed hard to understand. We have added the following text to the discussion.

Another explanation posits that homotypic zippers consisting solely of *cis*-homodimers are kinetically easier to form than heterotypic zippers since in a homotypic zipper, either “wing” of the new *cis* dimer can form *trans* interactions with the wing at the chain terminus. In contrast, in a hetero-dimeric zipper, only one wing can form homophilic interactions with the chain terminus (Figure 1D). A preference for homotypic zippers would then reduce the diversity required in the chain termination model since, in this model, it is essential that all isoforms be incorporated into a growing zipper. The formation of long homotypic zippers might lead to a repulsive phenotype even when mismatches are present.

Reviewer #2 (Recommendations for the authors):In two instances in the first page of the introduction the authors erroneously generalize to “vertebrates” the fact that cPcdhs are in three gene clusters. Fish do not have β clusters, but rather have 2 each of the α and γ clusters, and Xenopus tropicalis also lacks a β cluster (https://pubmed.ncbi.nlm.nih.gov/27261006/). “Mammals” would be an appropriate substitution.

We thank the reviewer for pointing out this mistake, we have changed the relevant instances to “mammalian”.

Regarding the presentation of the data in Figure 2, several of the isoforms show now interactions at all (α121-5, γA41-4, γB51-4). Later in the manuscript, AUC data are introduced that demonstrate that all three of these proteins do form homodimers. This would be worth mentioning on p. 8, as it alleviates potential worries about protein quality.

We have moved the introduction of the AUC data to the paragraph presenting the SPR experiments shown in Figure 2.

The result section on the γC41-4 structures is underwhelming. First, the RMSD comparisons and correlations to sequence similarity provide little additional useful information. Second, the attempt at an explanation of the low homodimerization affinity of γC41-4 does not seem to hold up upon inspection of the structures or Figure 3—figure supplement 2. The authors try to argue that the low affinity may be due to “two buried charges in the interface, E78 and D290.” E78 is in proximity to R89 and in some of the three structural instances forms a (longish) hydrogen bond to S344 across the interface, and other instances it is not quite buried. Similarly, D290 is hardly buried, and in proximity to K127 and R156 across the interface and H255 and H304 on the same subunit. Looking at the electrostatic surfaces of the interface similarly does not suggest a strong, if any, electrostatic mismatch.

We have removed some of the unnecessary RMSD and sequence identity tables as suggested and have conducted mutagenesis experiments to assess the role of E78 and D290 in γC4’s binding affinity. Mutagenesis of D290 to neutral amino acids did not increase γC4 EC1–4’s affinity in AUC experiments. However mutagenesis of E78 to either alanine and glutamine did significantly increase the binding affinity: Wild-type γC4 EC1–4 is essentially monomeric in AUC (>500 mM K_D_) while γC4 EC1–4 E78A has a KD of 58 mM and γC4 EC1–4 E78Q has a K_D_ of 83 mM. We also mutated the opposed residue to E78 in the γC4 interface to a basic amino acid to see if this compensated for the negative impact of E78. This was successful with γC4 EC1–4 S344R showing a K_D_ of 112 mM. These results are shown in Figure 3 and Figure 3—figure supplement 1 and discussed in the text.

The experiments, using mutagenesis, to investigate the asymmetry of cis dimerization are very interesting. It is a bit disappointing that the authors do not present a gain-of-function experiments – all the mutations are loss-of-function. Although the fact that the mutations lead to selective loss-of-function of specific interfaces do provide quite strong support for the inferences the authors make. One exception is this statement: “SPR data for the γB7 mutants over the αC2 surface suggests αC2 preferentially occupies the EC6-only side in αC2/γB7 dimers”. The experiment is a negative result, showing, using a Y-to-G mutation, that Y532 is not essential to the potential EC5-6 interface. So while the authors do wisely use the word “suggests”, the above caveat should be presented as well.

We have added this caveat to the text.

Can the authors speculate as to why the trans interactions have evolved to be so specific?

We believe that this is a requirement of the chain termination model. In the second paragraph of the discussion, line 500, we write:

“High fidelity homophilic interaction is a strict requirement of the chain termination model for the barcoding of vertebrate neurons and has been accomplished through the exploitation of a multidomain interface of almost 4000 Å2 (Nicoludis et al., 2019) that enables the positioning of enough “negative constraints” (Sergeeva et al., 2020) to preclude the dimerization of about 1600 heterophilic pairs of 58 mouse cPcdh isoforms (Rubinstein et al., 2017).”

On p. 16, lines 23-26, the sentence is confusing and should be rephrased. Are the authors suggesting that all cPcdhs will form dimers when constrained to a 2D membrane surface? If so, the use of the expression “monomeric cPcdhs” is confusing (i.e. do the authors mean cPcdhs that fail to dimerize in solution, or truly “monomeric” cPcdhs?).

By monomeric, we mean molecules which do not form measurable cis dimers in solution as now clarified in the text. Our assumption, based on our and others’ previous studies, is that cPcdhs need to form cis dimers in order to be efficiently delivered to the cell surface. This assumption explains the need for co-expression with “carrier” cPcdhs for cell surface delivery of α cPcdhs and cPcdh-gC4 (Thu et al., 2014). There are a number of cPcdhs which fail to cis homodimerize in solution which do not require co-expression with a carrier cPcdh for cell surface delivery, such as gA4 and gC3, and we assume that these molecules can form cis-homodimers in a membrane environment (Goodman et al., 2017). In line with this, whilst our SPR data clearly shows a wide range of cis binding strengths, we only conclude that interactions which fail to form in our experiments are comparatively weak not that they cannot form in a membrane environment.

Reviewer #3 (Recommendations for the authors):The authors may consider adding experiments that would extend the new conclusions drawn. One of the most interesting analyses is presented in Figure 3—figure supplement 2, where the structures of the C4 and B2 trans dimer interfaces are compared. It is suggested that C4 residue E78 may be destabilizing; could the authors continue their fruitful mutational approach here and determine if this is the case? The same could be done for D290 in C4. Mutating these residues or any adjacent ones so that they match those of gB2 or a delta2 Pcdh could be informative.

We have now conducted the mutagenesis experiments suggested to examine the contribution of particular interfacial residues to the weak gC4 trans interaction. These experiments indeed show that E78 (but not D290) negatively influences gC4 trans affinity with E78A and E78Q mutations both improving gC4 trans affinity in the AUC at least five-fold (Figure 3 / Figure 3—figure supplement 1).

Similarly, given what the authors have admirably shown regarding the mechanism of cis interaction and some of the residues that are crucial for the EC5/6 vs EC6 side of the dimer, can they utilize the data in Figure 4—figure supplement 3 to make predictions about EC5/6 vs EC6 “handedness” preference and test a few further examples to determine if this is likely to be generally predictable from sequence alone? If so then an interaction matrix could be determined that would help drive predictions in functional studies.

We did try to test further examples of side preferences based on our structure/sequence predictions. These cis interface mutations, which are listed on page 30 in the Materials and methods section, unfortunately tend to have a deleterious impact on protein expression in our system. Further exploration of this will therefore require a concerted effort into developing cPcdh protein expression methods which is beyond the scope of this paper.

The Discussion section could be improved by more clearly detailing how the new results here may explain, or are either consistent or inconsistent with, prior functional studies. For example, C3 does not homodimerize in cis (or at least does not do so preferentially). And yet in Lefebvre et al., 2012, re-introducing only C3 in a γ cluster knock out rescued self-avoidance in starburst amacrine cells while in Molumby et al., 2016 expression of C3 in cortical neurons on a knockout background rescued and actually enhanced dendrite arbor complexity. Similarly, alphaC2 is able to solely effect axonal tiling in some neurons (Chen et al., 2017). How do the authors envision the preferences for interfamily cis dimerization playing out functionally then? It is stated on the third page of the discussion (there are no page numbers), lines 6-10, that carrier isoforms may be able to fill both the EC6 and EC5/6 roles. If this is so then what would the importance of the interaction preferences so nicely shown here be? Some more exploration of this, perhaps with some predictions to test in future functional studies, would be most helpful. On that same page, final paragraph, it is noted that weak homophilic cis interactions may indicate that once in the membrane, isoforms re-sort to allow for heterodimeric zippers that are a prerequisite for the chain termination model. But again, if in several cases expression of a single isoform can rescue KO phenotypes, perhaps the model needs to be amended? It is worth exploring that more deeply here.

In situations where only a single cPcdh is expressed, zippers are expected to form and to initiate avoidance (an additional clarification has been added to the paper). aC2 alone is able to mediate tiling since tiling is achieved by neurites avoiding one another regardless of whether they originate from the same neuron or not. The failure of alternate α cPcdhs and the C-type gC4 to reach the cell membrane when expressed alone would suggest these molecules would be ineffectual when singly expressed but the remaining cPcdhs could operate singly.

Our assumption is that gC3 can form homophilic cis interactions on a membrane. The basis for this assumption is its efficient cell surface delivery when singly expressed and its ability to facilitate α cell surface delivery when co-expressed.

Similarly, C4 does appear to interact in trans similarly to other cPcdhs, so what might explain its special role in maintaining viability and neuronal survival? Do the authors see this as likely mediated by the unique cytoplasmic region, or does the weak trans interaction suggest possible cis interactions with non Pcdh molecules such as Ret or ROR or neuroligins, etc?

Since the γC4 trans interaction is structurally similar to other cPcdhs its isoform-specific biological functions cannot be directly attributed to its trans interaction. The weak affinity of γC4’s trans interaction could facilitate interaction with competing molecular partners but without knowing the precise binding sites or affinities of any putative interactions this is merely speculation. We have added further comments about γC4 to the discussion.

Some suggestions for figures:

The main figure 3 is very slight, with only panel A anything new (panel B is unnecessary). Data from Figure 3—figure supplement 1B and 2B should be moved into the main figure.

We have now added more data to figure 3 as suggested.

Figure 2—figure supplement 1: in panel B, the experiment with alphaC2 shown on top is not mentioned anywhere in the paper, nor is it explained in the legend. What is being tested in that experiment and what is the conclusion?

We thank the reviewer for drawing our attention to the lack of reference to the upper panel experiment shown in Figure 2—Figure supp. 1B. We have decided to remove this panel from the paper as the experiment does not add to the substance of this work.

In summary, this experiment was carried out to see whether a three domain (EC1–3) construct could bind to an immobilized EC1-4 construct. As shown in the figure aC2 EC1–3 can bind aC2 EC1–4 robustly, presumably via an EC1–3:EC2–4 antiparallel interaction. Previous work using chimeras in cell aggregation assays has previously indicated that three matching antiparallel EC domains is sufficient for trans interaction in at least some instances. However, since we only have data for a single cPcdh isoform no wider conclusions can be drawn and so we have removed the panel.

Figure 4B: is ND 'not determined' or 'not detected'? (if the latter, how does that differ from ">50"?). The legends in general are sparse and could be rewritten to explain the figures more.

We have replaced the acronym ND (none detected) with NB (no binding) in Figure 4B, to eliminate the confusion arising from dual meanings. An explanation now appears in the figure legend, defining NB as No Binding, which is different from K_D_ ">50" in that for interactions with K_D_s of >50 μM, there are detectable binding signals, but the interaction is too weak to accurately determine a K_D_. At the concentrations used in these SPR experiments (24-0.098μM), K_D_s greater than 50 μM cannot be accurately determined, since less than 30% of the binding sites are occupied hence introducing significant error in the calculations. An explanation has now been incorporated in the figure legend and the methods section to clearly define why some interactions have K_D_s >50 μM.

A few writing suggestions:

The Introduction section could be better organized. There is repetition of information between the first and third paragraphs. There are also quite a few sentences that have extra or missing words or are otherwise ungrammatical. For example, on page 2 of the introduction, lines 29-30; this sentence should be rewritten. Line 30 of the section titled "Clustered protocadherin cis interactions are promiscuous with a range of interaction strengths" has an extra word "which". There are other instances, please proofread for such grammatical errors/typos.

We have revised the introduction and reworded the specific cases mentioned above and we have proofread the entire document to catch and fix these errors. We have edited the sentence that in the current version appears on pg.5, lines 103-105, to read:

“…Rubinstein et al., (2015) proposed that cPcdhs located on apposed membrane surfaces would form an extended zipper-like lattice through alternating cis and trans interactions (Figure 1D).”

We have also appropriately deleted the extra “which” that was present on pg.11, line 357. A number of other grammatical errors have now been fixed.

SPR and AUC should be defined at first usage.

We have now defined these acronyms at the correct place.

The language used when explaining the preferential cis dimerization affinities for various cPcdhs could be more consistent or at least explained in more detail. While α Pcdhs are assumed to "only" be able to be an EC 5-6 donor, C3 is thought to "preferentially" be an EC5-6 donor. Since C3 does not dimerize in solution, wouldn't this indicate that C3 can "only" be a 5-6 EC donor? Is it known which other EC3-6 constructs can self-dimerize, and if so wouldn't that indicate which molecules can be considered "preferential" donors, while molecules which don't self-dimerize at all are "only" one kind of donor? Perhaps the authors could discuss.

We have now added further clarification to our interpretation of cis interaction side preferences to the discussion. The distinction between α cPcdhs and “carrier” cPcdhs (betas, gammas, aC2, gC3 and gC5) as obligate vs preferential EC5–6 or EC6-only donors is based on the assumption that cis dimerization is required for efficient cell surface delivery of cPcdhs. As such those molecules that cannot reach the cell surface when singly expressed are thought to be unable to form cis homodimers (alphas and gC4), whereas those that are efficiently delivered when expressed alone are thought to be able to homodimerize in cis. Co-expression experiments with α Pcdhs and various mutagenesis experiments support this model (Thu et al., 2014, Goodman et al., 2016b, and Goodman et al., 2017), but it hasn’t been conclusively shown biophysically. Our explanation for why some cPcdhs, such as gC3 and gA4, which are efficiently delivered to the cell surface do not form measurable cis dimers in our solution-based AUC experiments is that their cis dimer affinity is very weak (500 mM is the limit of our AUC experiments) but in the 2D environment of the cell membrane the apparent affinity for these membrane proximal interactions is increased. Any molecule which homodimerizes can by extension occupy both sides of the cis interaction. However, the absence of homodimerization does not necessarily imply that only one side is occupied; it is possible is just too low. We therefore use “preferential” rather than “only” to describe the cis binding characteristics for all carrier cPcdhs since this maintains consistency with the current body of cell-based data. Future work in a membrane environment is needed to show unambiguously that all carrier cPcdhs can form cis-homodimers.